# Host response during unresolved urinary tract infection alters female mammary tissue homeostasis through collagen deposition and TIMP1

Samantha Henry[1,2,8], Steven Macauley Lewis [1,2,8], Samantha Leeanne Cyrill [1,8], Mackenzie Kate Callaway[1], Deeptiman Chatterjee[1], Amritha Varshini Hanasoge Somasundara[1,3], Gina Jones[1], Xue-Yan He [4], Giuseppina Caligiuri[1], Michael Francis Ciccone[1], Isabella Andrea Diaz[1], Amelia Aumalika Biswas[1,5], Evelyn Hernandez[1], Taehoon Ha [1], John Erby Wilkinson[6], Mikala Egeblad [7], David Arthur Tuveson[1] & Camila Oresco dos Santos [1]✉

Exposure to pathogens throughout a lifetime influences immunity and organ function. Here, we explore how the systemic host-response to bacterial urinary tract infection (UTI) induces tissue-specific alterations to the mammary gland. Utilizing a combination of histological tissue analysis, single cell transcriptomics, and flow cytometry, we identify that mammary tissue from UTI-bearing mice displays collagen deposition, enlarged ductal structures, ductal hyperplasia with atypical epithelial transcriptomes and altered immune composition. Bacterial cells are absent in the mammary tissue and blood of UTI-bearing mice, therefore, alterations to the distal mammary tissue are mediated by the systemic host response to local infection. Furthermore, broad spectrum antibiotic treatment resolves the infection and restores mammary cellular and tissue homeostasis. Systemically, unresolved UTI correlates with increased plasma levels of the metalloproteinase inhibitor, TIMP1, which controls extracellular matrix remodeling and neutrophil function. Treatment of nulliparous and post-lactation UTI-bearing female mice with a TIMP1 neutralizing antibody, restores mammary tissue normal homeostasis, thus providing evidence for a link between the systemic host response during UTI and mammary gland alterations.

Alterations to the levels of whole-body circulating factors in response to perturbations, such as infections and hormonal level variations, can influence bodily function in lasting ways[1–4]. For example, while the onset of anemia may reduce the capacity of circulating oxygen, it also negatively alters organ functions by influencing cognitive function, chronic inflammation and immune responses to infection[5]. Similarly, bodily changes in response to persistent viral and bacterial infections, can result in pathological changes and excessive tissue damage, thus resulting in chronic diseases[4]. Specifically, systemic alterations to a female body in response to hormonal changes during puberty and pregnancy result in lasting changes to mammary epithelial cells (MECs) maturation, mammary, liver and lung function, and immunity,

---

thus representing a whole body changes that positively influence tissue homeostasis[6,7].

Associations between bacterial pathogenesis and breast biology are extensively studied in the context of mastitis, an infection process that has a negative impact on mammary health and lactation function[8]. However, little is known about how other whole-body responses induced by infections frequently experienced by women, such as those not local to the breast, affect breast health. One such concern that primarily afflicts women is urinary tract infections (UTIs), which affects one in three women globally[9]. UTIs elicit complex immunological host-responses as the body attempts to resolve the bacterial infection, by invoking local shedding of the urothelium, activation of resident innate immune cells, secretion of anti-microbial peptides, and systemic mobilization of innate and adaptive immune cells to the site of infection[10].

Here, we explore how the systemic host-response to bacterial urinary tract infection (UTI) induces tissue-specific alterations to mammary glands. By utilizing a well characterized, clinically relevant in vivo model of UTI in mice[11], we find that mammary tissue from UTI-bearing mice displays ductal hyperplasia, altered adipocyte content, and abnormal collagen accumulation. Uropathogenic e. coli bacterial cells (UPEC) are absent in the mammary tissue of UTI-bearing mice, suggesting that the observed tissue and cellular alterations are influenced by systemic host responses to infection, rather than localized infection. The onset of UTI also delays mammary regression after lactation, marked by the presence of residual milk protein and non-regressed ductal structures, thus indicating the disruption of canonical cellular dynamics that control mammary function and homeostasis. Single cell RNAsequencing (scRNA-seq) analysis illustrates that tissue alterations are accompanied by transcriptional changes in MECs, marked by upregulation of mechano-sensing programs. Such transcriptional alterations are associated with an altered communication between MECs and mammary fibroblasts, a finding that linked distal infections (UTI) with mammary ECM, stroma, and epithelial alterations. Probing the circulatory factors elevated in UTI-bearing mice, we identify the Tissue Inhibitor of Matrix Metalloproteinases 1 (TIMP1): a regulator of ECM remodeling and turnover that inhibits several matrix-degrading proteases (Matrix Metalloproteinases or MMPs)[12,13]. Preventing TIMP1 function with neutralizing antibody treatment during the course of infection fully restores mammary tissue homeostasis in UTI-bearing mice.

Collectively, our studies elucidate a mechanism by which a distal infection, commonly experienced by millions of women, can exert tissue-level effects beyond the local site, through modifying circulating factors. Our findings provide evidence for an expanded set of circulating factors, outside of hormones, that are induced by common life experiences and can have a profound effect on mammary biology and development, with relevance to potentially increasing the risk of future breast malignancy through collagen deposition driven changes to the stiffness of the tissue.

## Results

### Mammary tissue homeostasis is altered in mice with unresolved UTI

To study the impact of UTI on the mammary gland, we utilized a model system in which the bladders of female C57BL/6 mice were inoculated with the uropathogenic E. coli strain UTI89 (UPEC, referred to hereafter as UTI mice)[11], or PBS (control mice), via transurethral delivery (Supp. Fig. 1A). The onset of UTI was confirmed at 48 h post-infection (p.i.) with urine bacterial analysis (Supp. Fig. 1B, C). Mice that retained bacterial titers greater than $5 \times 10^4$ CFU/mL at 2 weeks p.i., and with signs of urothelial erosion, kidney immune infiltration, increased percentage of circulating granulocytes, and reduced percentage of circulating lymphocytes, were further analyzed (Supp. Fig. 1D–I). Infection retention phenotypes (urine

bacterial titers, urothelial erosion, kidney immune infiltration) were absent at 2 weeks p.i. timepoint in control PBS mice, and UTI mice treated with broad-spectrum antibiotic, Trimethoprim-Sulfamethoxazole (hereafter referred to as TMS), which were cleared of UPEC in the urine, as previously demonstrated (Supp. Fig. 1B–E)[14,15]. While the urine and bladder tissue were found positive for bacterial cells, analysis of total blood, plasma and mammary tissue from UTI-bearing mice failed to show bacteria, supporting that this is a model of localized bladder infection (Supp. Fig. 1J–L).

Analysis of mammary tissue from UTI-bearing mice revealed multiple mammary tissue abnormalities. H&E staining revealed expansion and enlargement of ductal structures in mammary tissue from UTI-bearing mice, which were 80% more frequent and 154% bigger than mammary structures from PBS mice (Fig. 1A–C). Our analysis also indicated increased adipogenesis and collagen content in the mammary tissue from UTI-bearing mice, defined by an 14% increase of Perilipin+ adipocytes, and an 84% increase of collagen content than mammary tissue from PBS mice (Fig. 1D–G). Further analysis of collagen content with Picrosirius Red staining supported that both thick and thin collagen fibers were at least 2-fold enriched in mammary tissue from UTI-bearing mice (Fig. 1H–J). Through analysis with two-photon microscopy and second harmonic generation (SHG) imaging, we found that the collagen present in the mammary gland of UTI-bearing mice displayed increased aligned architecture, suggesting that collagen accumulation could represent a combination of both altered synthesis/deposition and abnormal remodeling (Fig. 1K–M)[16]. Altered collagen deposition was not detected in other distal organs such as the pancreas, spleen, intestine, liver and lungs from UTI-bearing mice, suggesting that the ECM alterations resulting from unresolved UTI were specific in mammary gland tissue (Supp. Fig. 2A). We also found similar levels of corticosterone in the plasma of PBS control and UTI-bearing mice, suggesting that changes to this canonical regulator of infection responses and mammary tissue homeostasis does not represent the basis for UTI-associated changes to the gland (Supp. Fig. 2B).

An overall normalization of duct hyperplasia (197%), adipocyte counts (23%), and collagen content (41%) was observed in TMS treated animals, compared to non-TMS treated, UTI-bearing animals, suggesting that tissue-level alterations induced by UTI are caused by the sustained host response to infection (Fig. 1A–G). Furthermore, TMS treatment of PBS control, healthy animals did not alter mammary collagen content, indicating that antibiotic treatment alone does not induce changes to the gland (Supp. Fig. 2C–D). We also investigated whether a short term exposure to UTI symptoms would induce mammary tissue alterations. In doing so female C57BL/6 mice were inoculated with UTI89 uropathogenic strain (UTI mice), or PBS (control mice), via transurethral delivery, following UTI confirmation at 48 h p.i., and urine and mammary tissue analysis at 72 h p.i. (Supp. Fig. 2E). We found that mammary tissue from UTI-bearing mice compared to PBS controls at 72 h p.i. showed no increase in ductal size or collagen deposition in UTI mice, suggesting that mammary alterations require sustained, systemic host response during UTI (Supp. Fig. 2F–I). We also found similar plasma estrogen levels in UTI-bearing mice at 72 h or 2 weeks p.i., further excluding abnormal levels of a master regulators of mammary tissue development as the basis for UTI-associated changes to the gland, and supporting that mammary tissue changes in response to the onset of UTI are likely to be regulated by sustained responses to UTI (Supp. Fig. 2J).

### The onset of UTI induces cellular and molecular alterations to mammary tissue

Changes to mammary stroma composition have been linked with alterations to MEC lineage identity and gene regulation[17–19]. Therefore, we next employed scRNA-seq analysis to determine whether changes

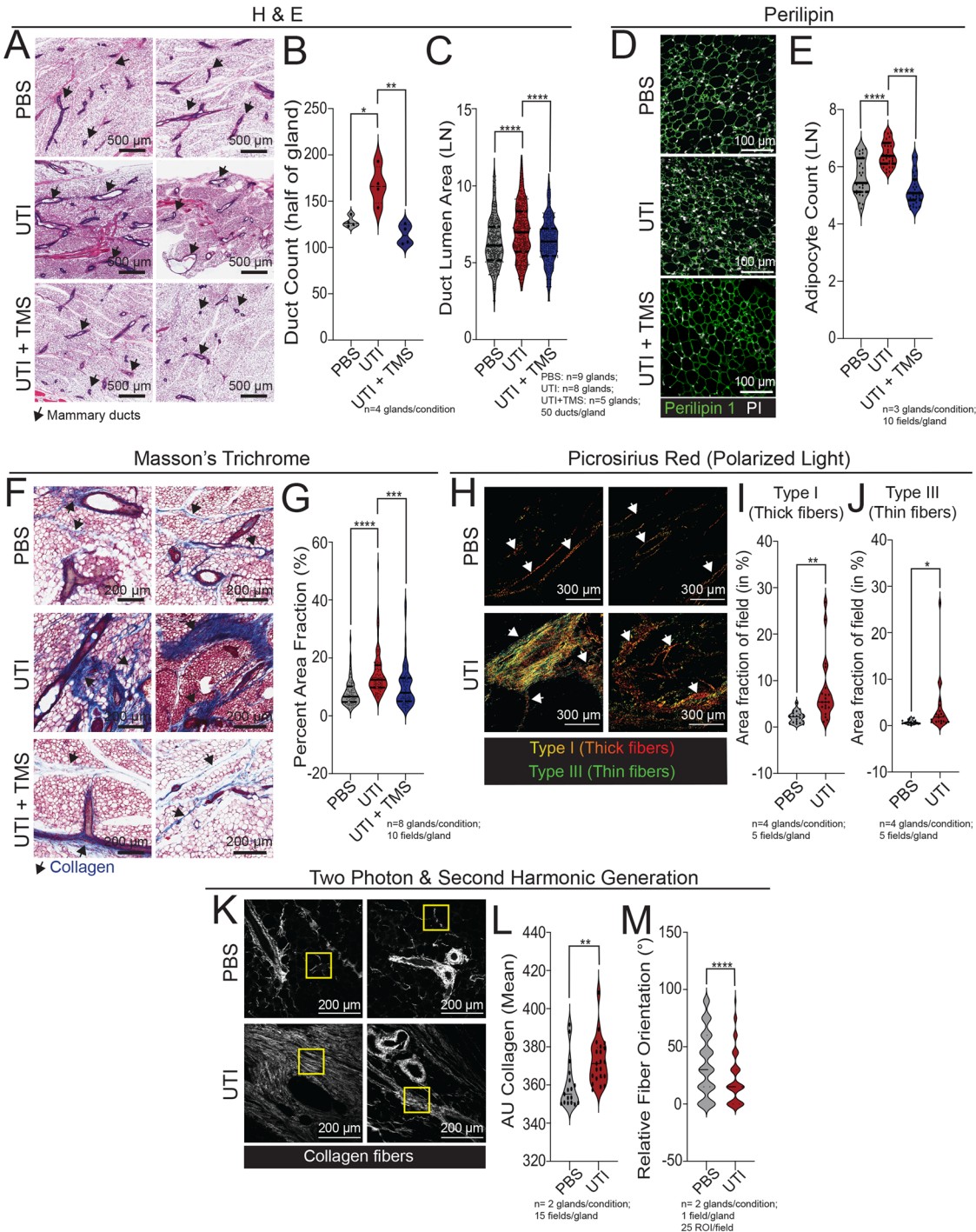

induced to mammary tissue in UTI-bearing mice resulted in alterations to epithelial cell abundance or transcriptional activity. In doing so, we generated scRNA-seq profiles of total mammary cells from UTI-bearing mice (2 weeks p.i.), and compared it to our previously published dataset generated from healthy, nulliparous mice (hereafter referred to as NP)[20].

Hierarchical clustering identified all 3 major epithelial lineages in mammary tissue from both healthy and UTI-bearing mice, including Basal-Myoepithelial cells (BM), Luminal adaptive secretory precursor cells (LASP), and Luminal hormone-sensing cells (LHS), with no statistically significant changes to cell abundance across health and UTI conditions (Fig. 2A, B)[21]. Interestingly, our analysis indicated the enrichment of MEC-ECM communication and mechano-sensing pathways in MECs from UTI-bearing mice (Supp. Fig. 3A, B). In fact, YAP

networks, known canonical transducers and coordinators of transcription in response to mechanical signals, such as from ECM deposition, were also found to be enriched in MECs from UTI-bearing mice, thus supporting alterations to MEC transcriptional states that associate with collagen deposition increase in response to UTI (Fig. 2C).

In the mammary gland, collagen remodeling is controlled throughout all major developmental stages, with fibroblasts playing a primary role in collagen synthesis, remodeling, and deposition[22]. Therefore, we next utilized our scRNA-seq datasets to investigate whether fibroblast abundance or inferred functions could be influencing MECs in UTI-bearing mice. In doing so, we performed a deep-cluster approach on cells expressing the stromal marker Sparc and lacking the expression of pan-epithelial marker (Epcam), pan-

**Fig. 1 | UTI-bearing mice have altered mammary tissue homeostasis. A–C** H&E staining of mammary tissue sections and (**B**) quantification of ductal counts (PBS vs UTI: *p = 0.0274, UTI + TMS vs UTI: **p = 0.0077) and (**C**) quantification of ductal lumen area in mammary tissue (PBS vs UTI: ****p < 0.0001, UTI + TMS vs UTI: ****p < 0.0001) from PBS control mice (n = 9), from UTI-bearing mice (n = 8) and from UTI-bearing mice treated with TMS (n = 5) at 2 weeks post-infection (p.i.). Scale bar = 500 µm. Arrows indicate mammary ducts. Violin plots show data distribution with median (solid line) and quartiles (dashed line) indicated. Statistical analysis was conducted with one-way ANOVA with Tukey's multiple correction. **D, E** Immunostaining of mammary tissue sections and (**E**) quantification of Perilipin+ adipocytes (green) and PI (white) from PBS control mice (n = 3), from UTI-bearing mice (n = 3) and from UTI-bearing mice treated with TMS (n = 3) at 2 weeks post-infection (p.i.). Scale bar = 100 µm. Violin plots show data distribution with median (solid line) and quartiles (dashed line) indicated. Statistical analysis was conducted with one-way ANOVA with Tukey's multiple correction; PBS vs UTI: ****p < 0.0001, UTI + TMS vs UTI: ****p < 0.0001. **F, G** Masson's trichrome staining of mammary tissue sections and (**G**) quantification of positively stained collagen (in blue) from PBS control mice (n = 8), from UTI-bearing mice (n = 8), and from UTI-bearing mice treated with TMS (n = 8) at 2 weeks p.i. Scale bar = 200 µm. Arrows indicate positively stained collagen (in blue). Violin plots show data distribution with median (solid line) and quartiles (dashed line) indicated. Statistical analysis was conducted with one-way ANOVA with Tukey's multiple correction; PBS vs UTI: ****p < 0.0001, UTI + TMS vs UTI: ***p < 0.0006. **H–J** Picrosirius red staining in mammary tissue sections, and quantification of (**I**) type I collagen (thick fibers, **p = 0.0018) and (**J**) type III collagen (thin fibers, *p = 0.0315), from PBS control mice (n = 4), and from UTI-bearing mice (n = 4), at 2 weeks p.i. Stained slides were imaged under linearly polarized light. Scale bar = 300 µm. Arrows indicate collagen fibers. Violin plots show data distribution with median (solid line) and quartiles (dashed line) indicated. Statistical analysis was conducted with an unpaired t-test with Welch's correction. **K–M** Two-photon & second harmonic generation imaging of H&E stained mammary gland slides, and quantification of (**L**) total collagen signal (**p = 0.0042), and (**M**) collagen orientation (****p < 0.0001), from PBS control mice (n = 2), and from UTI-bearing mice (n = 2), at 2 weeks p.i. Stained slides were imaged under linearly polarized light. Scale bar = 200 µm. Yellow boxes indicate collagen fibers. Violin plots show data distribution with median (solid line) and quartiles (dashed line) indicated. Statistical analysis was conducted with an unpaired t-test with Welch's correction. Source data and p-values are provided as a Source Data file.

endothelial marker (Pecam-1) and pan-immune marker (Ptprc), as previously described[23,24]. Such approach identified 3 populations of fibroblasts (referred hereafter as Nulliparous fibroblasts, NPF), from which cluster NPF2 was defined to be significantly enriched in samples from UTI-bearing mice (Supp. Fig. 3C, D). Differential expression analysis of gene markers indicated that fibroblasts enriched in mammary tissue from UTI-bearing mice (NPF2) are defined by high expression of genes that are associated with ECM/collagen remodeling such as *Myoc, Ogn, Hmcn2*, and *Thbs4* (Supp. Fig. E)[25–28]. In agreement, additional analysis of fibroblast lineage state indicated that NPF2 cells showed signatures of fibroblasts described to have a higher collagen synthesis potential, thus further supporting that an ongoing UTI stimulates the presence of fibroblasts with collagen related functions in the mammary gland (Fig. 2D)[29]. This was in marked contrast to fibroblasts subtypes also present in mammary tissue from healthy animals, which were classified as immunoregulatory-like states (NPF1), and myofibroblasts (NPF3) (Fig. 2D). In fact, immune staining analysis of Fibronectin levels, an extracellular matrix protein that interacts with collagen, and highly expressed in NPF2 fibroblasts, confirmed increased abundance of Fn1+ stromal cells around mammary gland structures of UTI-bearing mice, thus further supporting that an ongoing UTI stimulates the presence of fibroblasts with collagen related functions in the mammary gland (Fig. 2D, E, red asterixis). Collectively, our analysis suggests that structural changes (collagen deposition) to mammary tissue in UTI-bearing mice are influencing the transcriptional state of MECs, and the abundance of collagen-associated cell types (fibroblasts).

We next asked whether we could infer a direct communication between MECs and fibroblasts in mammary tissue from UTI-bearing mice, using CellChat analysis of scRNA-seq datasets from the mammary gland of control and UTI-bearing mice[30]. We found stronger emission of ECM related signals, specifically collagen and laminin ones, by NPF2 fibroblasts present in the mammary tissue of UTI-bearing mice to major MEC subtypes, compared to signal sent by NPF1 fibroblasts, which is present in tissue from both healthy and UTI-bearing mice (Fig. 2F, G, Supp. Fig. 3F, G). Further analysis of receptor-ligand pairs predicted that Itga9-Itgb1 pairing was predominant between NPF1 and NPF2 fibroblasts, and major MEC subtypes in the mammary tissue of UTI-bearing mice, while Itgav-Itga8 pairing was more frequent between NPF1 fibroblasts and MECs in the non-UTI mammary tissue (Fig. 2H, I). Interestingly, Itga3-Itgb1 pairing, which was predicted to be in place across conditions between fibroblasts and BM cells, was also more frequent between fibroblasts and LHS cell types in UTI-bearing mice (Fig. 2H, I). Collectively, this analysis provides in silico support of signaling mechanisms associated with specific subtypes of fibroblasts and receptor-ligand pairing that are altered in UTI-bearing mice, and associate with the increased mammary collagen accumulation.

### UTI interferes with mammary gland involution post-lactation

Post-partum women are at increased risk to develop recurrent and difficult to treat UTIs[31]. Having characterized the effects of unresolved UTI on mammary tissue homeostasis in nulliparous mice, we next investigated whether a critical timepoint of mammary gland development, the resolution of pregnancy-induced changes that occur during post-partum tissue involution, would be impacted in UTI-bearing mice. In doing so, we analyzed the mammary tissue of actively nursing female mice (full-term pregnancy, 10 days of nursing) post transurethral delivery of either UTI89 uropathogenic bacteria (UTI condition) or PBS control. After confirmation of bacteriuria, 48 h p.i., the offspring were weaned to induce forced mammary involution. Bacteriuria and tissue analysis were conducted 10 days post-lactation involution (referred hereafter as PLI), a time-point at which the mammary gland is expected to have significantly regressed to its pre-pregnancy state (Supp. Fig. 4A)[32]. UTI-bearing PLI mice 2 weeks p.i. showed bacteria titers, bladder and kidney alterations similarly to the ones observed in nulliparous mice, relative to PBS controls, thus indicating the onset of UTI and associated phenotypes. (Supp. Fig. 4B, C).

Analysis of mammary tissue from UTI-bearing PLI mice revealed mammary tissue abnormalities. H&E staining revealed increased number of ductal structures in the mammary tissue of UTI-bearing PLI mice (black arrows), which we confirmed with immunofluorescent image analysis, and indicated a 76% increase on the number of ductal structures (defined by Cytokeratin 5 staining) (Fig. 3A-C). H&E tissue analysis of mammary tissue from UTI-bearing PLI mice also suggested increased milk accumulation into the ducts, which we investigated with the quantification of the milk-associated protein β-casein, an analysis that reveled a 733% increase of β-casein signal, relative to PBS control, suggesting that unresolved UTI impacts on the post-lactation involution process of the mammary gland (Fig. 3D).

We also identified that mammary tissue from UTI-bearing PLI mice displayed a statistically significant, 24% increase in collagen content, compared to PBS control, an alteration also present in the tissue of nulliparous (never pregnant) mice, thus suggesting collagen alterations that are brought by the onset of UTI to mammary gland, and are independent of parity (Fig. 3E, F, Supp. Fig. 4D). Analysis of adipocyte content, a cell type that is reduced in the

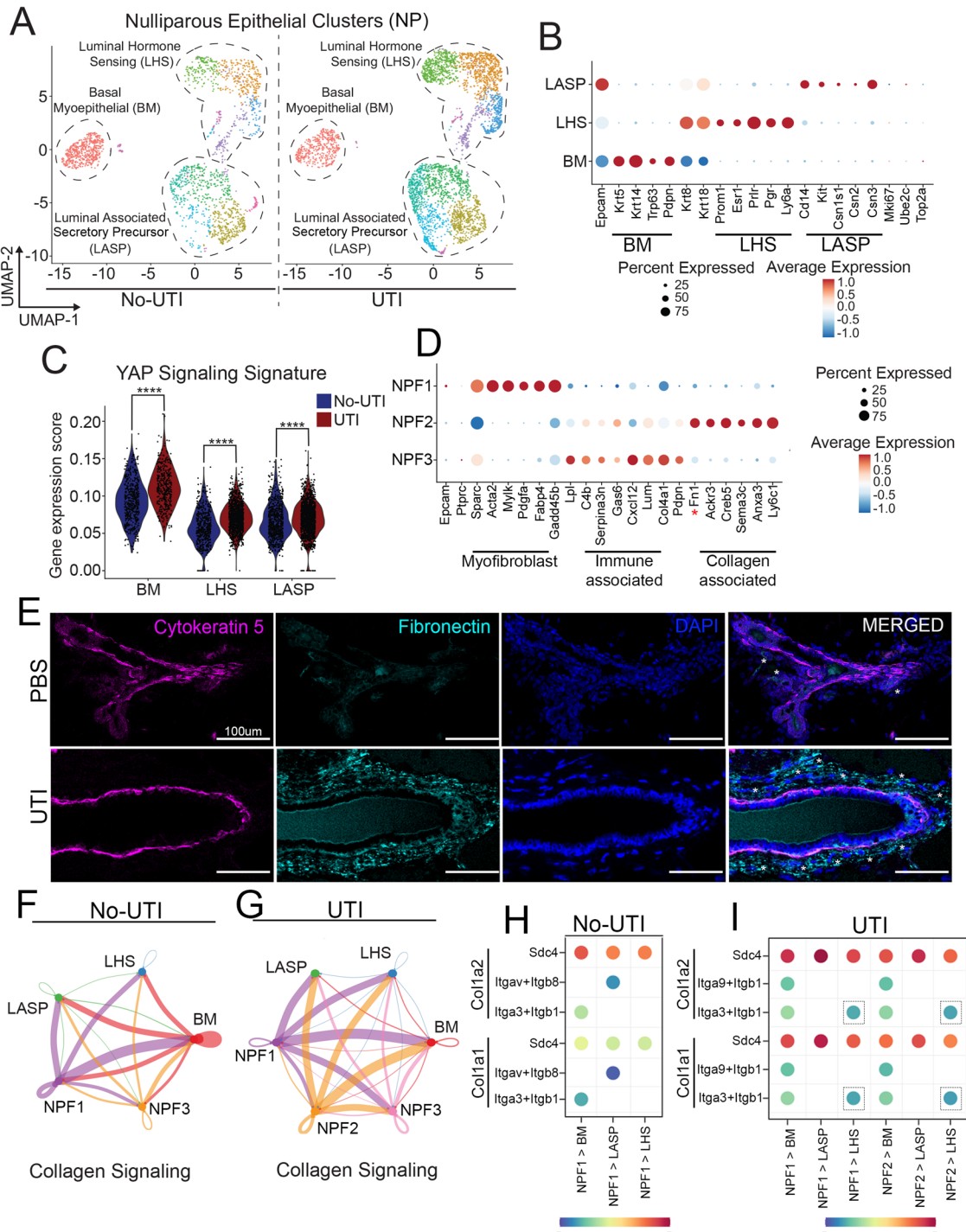

**Fig. 2 | scRNA-seq analysis identifies mammary molecular and cellular alterations induced by the onset of UTI. A, B** UMAP and (**B**) lineage classification of mammary epithelial cells from uninfected mice (No-UTI, $n = 2$) and UTI-bearing mice (UTI, 2 weeks post-infection (p.i.), $n = 2$). **C** Violin plot showing gene expression score of YAP Signaling signature across major epithelial lineages from uninfected mice (No-UTI, blue) and UTI-bearing mice (red). Statistical analysis was conducted with a non-parametric Wilcox test; BM: ****$p < 2.22e-16$, LHS: ****$p < 2.22e-16$, LASP: ****$p < 2.22e-16$. **D** Dotplot showing average expression of genes associated with myofibroblast state, immuno-related fibroblast state, or collagen-Fn1-related fibroblast. **E** Immunostaining analysis showing Cytokeratin 5+ MECs (magenta), DAPI (blue), and Fibronectin (Fn1)+ fibroblasts (cyan) in mammary tissue from PBS control and UTI-bearing mice, at 2 weeks p.i. Scale bar = 200 μm. White asterixis (*) indicate Fn1+ fibroblasts. Representative images shown from $n = 2$ PBS and $n = 2$ UTI-bearing mice (**F-G**) Intensity plot showing Collagen signaling cell-cell interactions in fibroblasts from (**F**) uninfected mice (No-UTI) and (**G**) UTI-bearing mice (UTI). (**H-I**). Top significant ligand receptor pairs associated with Collagen signaling in (**H**) uninfected mice (No-UTI) and (**I**) UTI-bearing mice (UTI). Dashed squares indicate gains of pairs in LHS cells from UTI-bearing mice, at 2 weeks p.i.

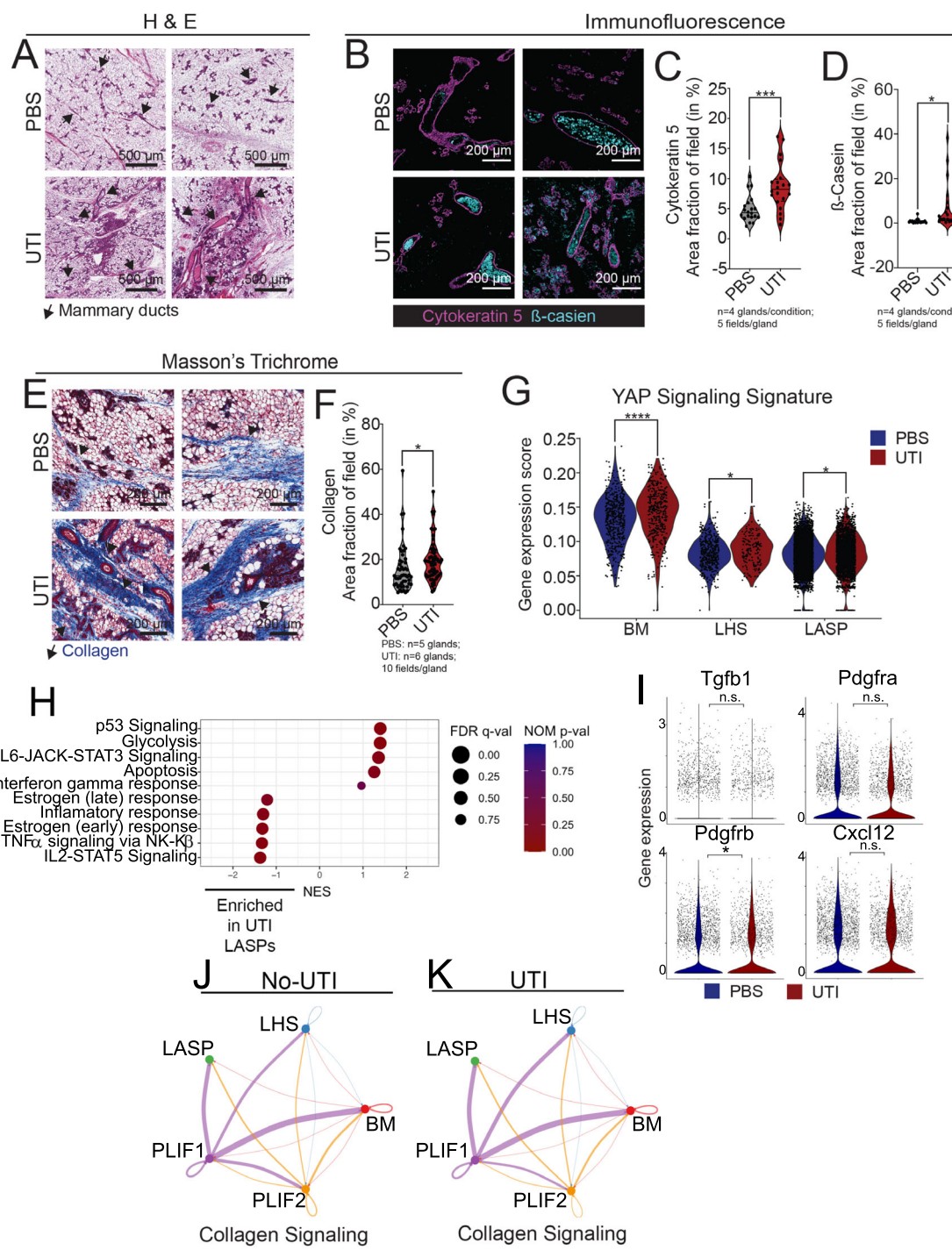

**Fig. 3 | The onset of UTI alters mammary gland post-lactation involution (PLI).**
**A** H&E staining of mammary tissue sections from PLI PBS control mice and PLI UTI-bearing mice, at 2 weeks post-infection (p.i.). Representative images shown from $n = 5$ PBS and $n = 7$ UTI-bearing post-lactation involution mice. Arrows indicate mammary ducts. **B**–**D** Immunostaining analysis showing Cytokeratin 5+ MECs (magenta) and β-casein (cyan), and quantification of (**C**) duct structures (inferred by Cytokeratin 5 signal, ***$p = 0.0009$), and (**D**) β-casein levels (*$p = 0.0153$) in the mammary tissue from PLI PBS control mice ($n = 4$) and PLI UTI-bearing mice ($n = 4$), at 2 weeks p.i. Scale bar = 200 μm. Violin plots show data distribution with median (solid line) and quartiles (dashed line) indicated. Statistical analysis was conducted with an unpaired t-test with Welch's correction. **E, F** Masson's trichrome staining of mammary tissue sections and (**F**) quantification of positively stained collagen (in blue) from PLI PBS control mice ($n = 5$) and PLI UTI-bearing mice ($n = 6$), at 2 weeks p.i. Scale bar = 200 μm. Arrows indicate positively stained collagen (in blue). Violin plots show data distribution with median (solid line) and quartiles (dashed line)

indicated. Statistical analysis was conducted with an unpaired t-test with Welch's correction; *$p = 0.0474$. **G** Violin plot showing gene expression score of YAP Signaling signature across major epithelial lineages from PLI PBS control mice (blue, $n = 2$) and PLI UTI-bearing mice (red, $n = 2$), at 2 weeks p.i. Statistical analysis was conducted with a non-parametric Wilcox test; BM: ****$p = 2.80e\text{-}06$, LHS: *$p = 0.028$, LASP: *$p = 0.021$. **H** Gene-set enrichment analysis (GSEA) of LASP cells from PLI PBS control mice (positive NES) and PLI UTI-bearing mice (negative NES), at 2 weeks p.i. Plot displays the top 5 pathways for each condition. **I** Violin plots showing gene expression levels of involution-induced, activated fibroblasts. Statistical analysis was conducted with a non-parametric Wilcox test; *Tgfb1* mRNA levels: $p = 0.27$, *Pdgfra* mRNA levels: $p = 0.078$, *Pdgfrb* mRNA levels: *$p = 0.044$, *Cxcl12* mRNA levels: $p = 0.76$. Intensity plot showing Collagen signaling cell-cell interactions in fibroblasts from (**J**) PLI PBS control mice and (**K**) PLI UTI-bearing mice at 2 weeks p.i. Source data and p-values are provided as a Source Data file.

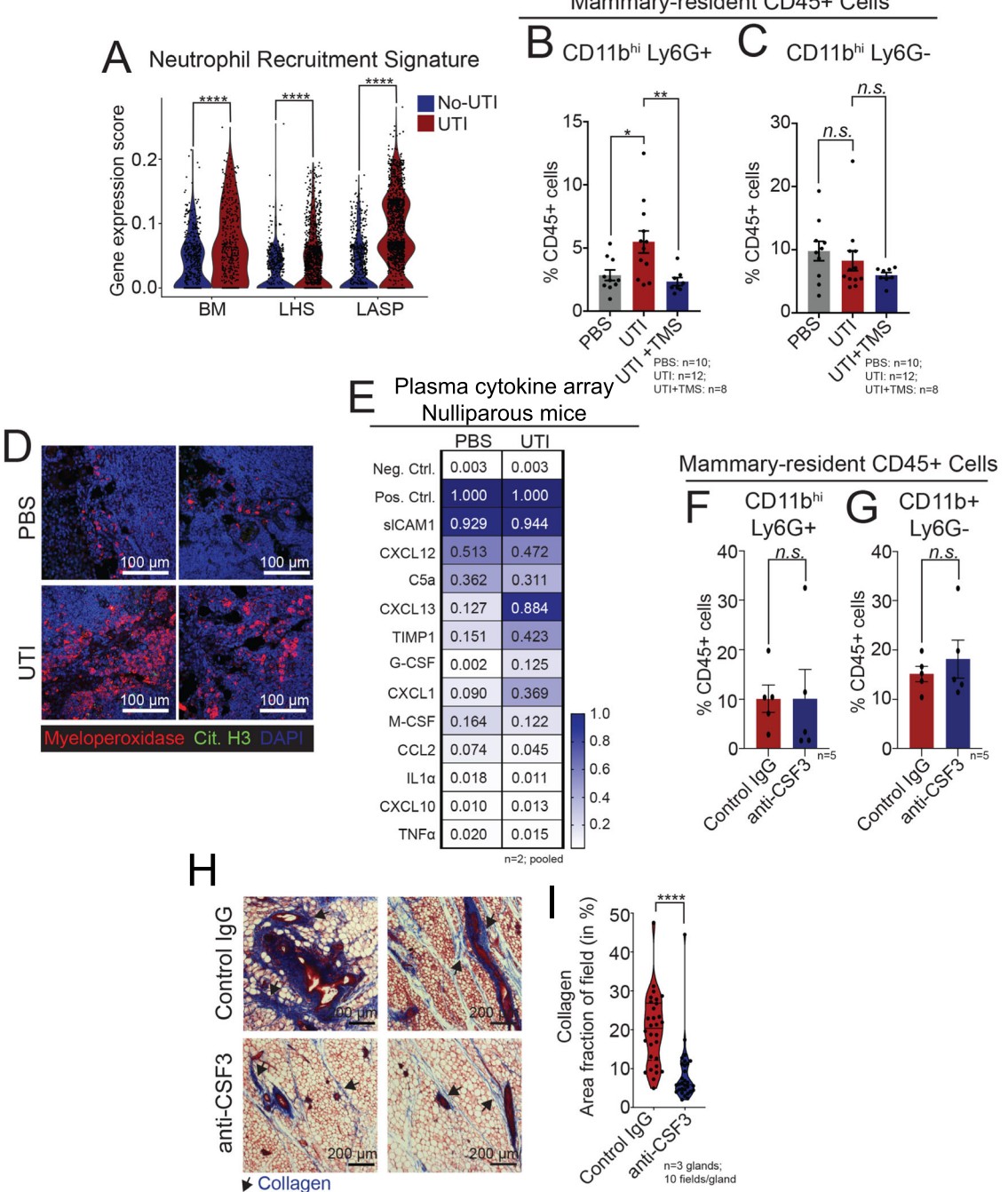

**Fig. 4 | UTI-bearing mice have altered mammary neutrophil infiltration and elevated G-CSF in circulating plasma. A** Violin plot showing gene expression score of Neutrophil Recruitment signature across major epithelial lineages from nulliparous (NP) PBS control mice (blue) and NP UTI-bearing mice (red). Statistical analysis was conducted with a non-parametric Wilcox test; BM: ****$p$ = 2.30e-11, LHS: ****$p$ = 1.50e-11, LASP: ****$p$ < 2.22e-16. Flow cytometry quantification of mammary-resident (**B**) CD11b$^{hi}$Ly6G+ neutrophils (PBS vs UTI: *$p$ = 0.02, UTI + TMS vs UTI: **$p$ = 0.009), and (**C**) CD11b + Ly6G- (myeloid, PBS vs UTI: $p$ = 0.7145, UTI + TMS vs UTI: $p$ = 0.5151) cells from NP PBS mice ($n$ = 10), from NP UTI-bearing mice ($n$ = 12), and from NP UTI-bearing mice treated with TMS ($n$ = 8) at 2 weeks post-infection (p.i.). Error bars represent mean ± S.E.M. Statistical analysis was conducted with one-way ANOVA with Tukey's multiple correction. **D** Immunostaining analysis of Myeloperoxidase+ neutrophils (Mpo + , red) and Citrullinated histone H3+ neutrophils (Cit. H3., in green) in mammary tissue sections from NP PBS control mice and NP UTI-bearing mice, at 2 weeks p.i. Representative images shown from $n$ = 4

PBS and $n$ = 6 UTI. Scale bar = 100 μm. **E** Cytokine quantification of plasma collected from NP PBS control mice ($n$ = 2 samples pooled) and NP UTI-bearing mice ($n$ = 2 samples pooled) at 2 weeks p.i. Flow cytometry quantification of mammary-resident (**F**) CD11b$^{hi}$Ly6G+ neutrophils ($p$ = 0.9969), and (**G**) CD11b + Ly6G- (myeloid, $p$ = 0.5017) cells from NP UTI-bearing mice after 6 doses of IgG control ($n$ = 5) or anti-CSF3 neutralizing antibody ($n$ = 5). Error bars represent mean ± S.E.M. Statistical analysis was conducted with one-way ANOVA with Tukey's multiple correction. **H, I** Masson's trichrome staining of mammary tissue sections and (**I**) quantification of positively stained collagen (in blue) from NP UTI-bearing mice after 6 doses of IgG control ($n$ = 3) or anti-CSF3 neutralizing antibody ($n$ = 3). Scale bar = 200 μm. Arrows indicate positively stained collagen (in blue). Violin plots show data distribution with median (solid line) and quartiles (dashed line) indicated. Statistical analysis was conducted with an unpaired t-test with Welch's correction; ****$p$ < 0.0001. Source data and p-values are provided as a Source Data file.

mammary tissue during pregnancy[33], but is reestablished early during post-lactation involution, showed a subtle, but non-statistically significant, cellular expansion in UTI-bearing PLI mice (Supp. Fig. 4E, F). Collectively, these findings suggest that during active tissue remodeling phase, as observed during the transition from lactation to involution, UTI-induced mammary alterations result in milk accumulation and collagen deposition.

We next performed scRNA-seq analysis, to further investigate the impact of an ongoing UTI on cellular and molecular states of mammary tissue during post-lactation involution. Our analysis identified all major MEC lineages (BM, LASP, and LHS), which were equally distributed in abundance across UTI and PBS conditions (Supp. Fig. 5A, B). We found that pathways linked with YAP-signaling signatures were also enriched in several MECs lineages from UTI-bearing mice, once again illustrating changes that influenced both never pregnant and involuting mammary tissue (Fig. 3G). Differential transcriptional analysis indicated that pathways associated with the transition between lactation and early steps of involution, such as IL2-STAT5 and TNFα signaling, were enriched in LASPs from PLI UTI-bearing mice, suggesting a yet to be fully engaged involution process (Fig. 3H). Interestingly, some of genes associated with such pathways were up-regulated across all MECs, thus suggesting that a signature of partial involution initiation was shared by all epithelial cells in UTI-bearing mice (Supp. Fig. 5C). In contrast, LASPs from PLI PBS control animals, were enriched with processes linked with active process of involution and immune recruitment, such as IL6/STAT3 and IFNγ responses, suggesting a molecular signature of ongoing involution (Fig. 3H). Collectively, these observations suggest that molecular processes associated with mammary involution were not fully activated in LASPs from PLI UTI-bearing mice, a finding that supports the observation of milk protein residue accumulation, and further suggests that the onset of UTI delays the post-lactation involution of mammary tissue.

The balance between YAP-signaling, collagen content, milk production is of extreme importance during lactation and involution. While collagen provides the environment for expansion and specialization of milk producing cells (LASPs) and contractile cells (BMs), the increased collagen provides the orientation for tissue wound healing during removal of dead cells during involution[34,35]. Such increased collagen concentration during involution is induced by specific subtypes of fibroblasts[29]. Our analysis of nulliparous tissue indicated that collagen and MEC transcriptional alterations in UTI-bearing mice were associated with specific subtypes of fibroblasts (Fig. 2E–H). Therefore, we next investigated whether specific populations of fibroblasts were more abundantly present in the mammary tissue from UTI-bearing PLI mice.

Through this analysis, we identified 2 fibroblast clusters (referred here after as post-lactation fibroblasts, PLIF), which we defined as distinct states of mammary fibroblasts according to the enrichment of collagen remodeling genes (PLF1) and myofibroblasts associated genes (PFLI2) (Supp. Fig. 5D, E). Unlike the observation of nulliparous UTI-bearing mice, we did not detect changes to the fibroblast abundance comparing UTI and PBS conditions, nor identified a nulliparous-like Fn1+ fibroblast signature, suggesting during post-lactation involution such fibroblast state is less common, and that the onset of UTI did not majorly alter already in place fibroblast identities (Supp. Fig. 5F). We also found that fibroblasts of both PBS control and UTI-bearing PLI mice expressed similar levels of fibroblast activated genes, suggesting that UTI processes did not perturb signatures associated with those induced by involution (Fig. 3I)[35]. Prediction of collagen signaling, and overall communication between fibroblasts and MECs in PLI animals, indicated an overall overlap of signals being sent by fibroblasts to MECs across UTI-bearing and PBS conditions (Fig. 3J, K, Supp. Fig. 5G, H). Taken together, these findings support that UTI-induced alteration to the

mammary tissue during post-lactation involution are not as prominent as the ones encountered in nulliparous animals.

## UTI induces infiltration of neutrophils into the mammary gland independent of systemic CSF3

UTIs elicit complex immunological host-responses, with the recruitment of specific immune cells, both locally to the bladder and systemically, as the body attempts to fight the bacterial infection[36,37]. Our initial findings indicate a sustained elevation of granulocytes in the circulation of UTI-bearing mice, as one of the symptoms that persisted 2 weeks p.i. (Supp. Fig. 1). In fact, and in addition to programs activated by collagen and ECM, our scRNA-seq datasets indicated that MECs from nulliparous UTI-bearing mice were also enriched for genes linked with neutrophil recruitment, suggesting that the elevated levels of neutrophils in the blood could be sensed by the mammary tissue (Fig. 4A). Interestingly, neutrophil recruitment signature was down-regulated in MECs from UTI-bearing PLI mice, suggesting that such mechanism is differentially modulated according to the state of mammary gland development (Supp. Fig. 6A). We therefore, set out to validate whether neutrophils were indeed recruited to the mammary gland in response to an ongoing UTI.

Flow cytometry analysis indicated a 93% increase in the abundance of total CD45+, CD11b^hiLy6G+ neutrophils in the mammary glands from UTI-bearing mice, compared to PBS mice, a population of immune cells that has been described to be expanded in order to attenuate inflammatory responses (Fig. 4B)[38]. In contrast, we did not detect changes to the abundance of CD11b^hiLy6G- immature-like myeloid cells, also known to increase in abundance in response to infections, in the mammary tissue of UTI-bearing and PBS animals (Fig. 4C)[39]. Neutrophils recruited to mammary tissue returned to normal levels upon treatment of UTI-bearing mice with TMS, suggesting that neutrophil expansion in the mammary tissue is driven by changes triggered during ongoing UTI (Fig. 4B, C). Analysis of an earlier time-point during the course of UTI (72 h p.i.) showed no significant difference in the abundance of neutrophils in mammary tissue, suggesting that the alterations observed in the gland at 2 weeks p.i. were a consequence of the sustained responses to UTI (Supp. Fig. 6B–D). We also did not find differences in the population of neutrophils across UTI and PBS conditions at 2 weeks p.i. in PLI mice, a finding that agrees with the levels of neutrophil recruitment signature detected in MECs (Supp. Fig. 6E).

To understand if neutrophil recruitment during UTI was mammary-specific, we evaluated multiple tissues across the body at 2 weeks p.i. In doing so, we utilized immunofluorescence staining to detect the neutrophil marker, myeloperoxidase (Mpo + ). This analysis indicated Mpo+ cells in the mammary tissue of UTI-bearing mice, supporting the flow cytometry expansion of this population of cells (Fig. 4D, red label). We also found that Mpo+ neutrophils in the mammary of UTI-bearing mice did not express high levels of citrullinated histone H3 (cit. H3), suggesting the lack of neutrophil extracellular traps (NETs) as a consequence of neutrophil recruitment to the mammary gland during UTI (Fig. 4D, green label)[40]. Additional analysis indicated increased Mpo+ neutrophils in the spleen, pancreas, lung and liver of UTI-bearing mice, with no statistically significant differences in the bone marrow and intestines comparing UTI to PBS conditions (Supp. Fig. 6F, G). This finding suggests that the neutrophil infiltration varies across tissues in UTI-bearing mice, and it is not exclusive to mammary tissue.

Given that increased levels of granulocytes are part of the systemic response to a UTI (Supp. Fig. 1), we sought to identify the circulatory factors regulating the mammary-specific phenotypes in UTI-bearing mice[41]. Unbiased analysis of plasma cytokines identified elevated levels of Granulocyte colony stimulation factor (G-CSF, 62-fold) and Chemokine (C-X-C) motif ligand 1 (CXCL1, 4-fold), known granulocyte-associated factors in the plasma of UTI-bearing mice

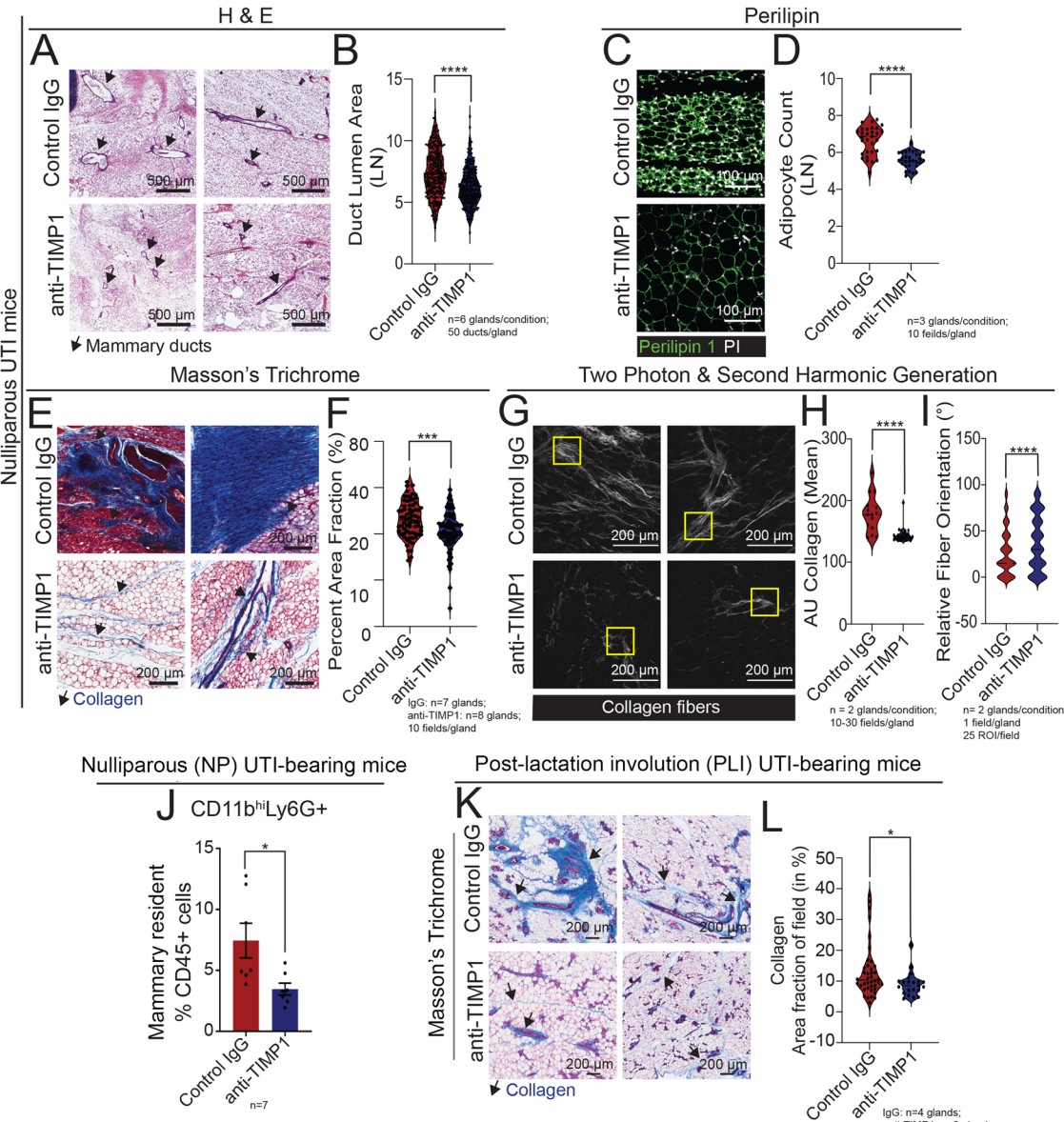

**Fig. 5 | Neutralization of TIMP1 normalizes mammary duct abnormality, ECM content, and neutrophil infiltration in nulliparous UTI-bearing mice. A, B** H&E staining of mammary tissue sections and (**B**) quantification of ductal lumen area in mammary tissue from nulliparous (NP) UTI-bearing mice after 6 doses of IgG control ($n = 6$) and anti-TIMP1 neutralizing antibody ($n = 6$). Scale bar = 500 μm. Arrows indicate mammary ducts. Violin plots show data distribution with median (solid line) and quartiles (dashed line) indicated. Statistical analysis was conducted with an unpaired t-test with Welch's correction; ****$p < 0.0001$. **C, D** Immunostaining of mammary tissue sections and (D) quantification of Perilipin+ adipocytes (green), and PI (white), from NP UTI-bearing mice after 6 doses of IgG control ($n = 3$) and anti-TIMP1 neutralizing antibody ($n = 3$). Scale bar = 100 μm. Violin plots show data distribution with median (solid line) and quartiles (dashed line) indicated. Statistical analysis was conducted with an unpaired t-test with Welch's correction; ****$p < 0.0001$. **E, F** Masson's trichrome staining of mammary tissue sections and (**F**) quantification of positively stained collagen (in blue) from NP UTI-bearing mice after 6 doses of IgG control ($n = 7$) or anti-TIMP1 neutralizing antibody ($n = 8$). Scale bar = 200 μm. Arrows indicate positively stained collagen (in blue). Violin plots show data distribution with median (solid line) and quartiles (dashed line) indicated. Statistical analysis was

conducted with an unpaired t-test with Welch's correction; ***$p = 0.0003$. **G–I** Two-photon & second harmonic generation imaging of H&E stained mammary gland slides, and quantification of (**H**) total collagen signal (****$p < 0.0001$), and (**I**) collagen orientation (****$p < 0.0001$) from NP UTI-bearing mice after 6 doses of IgG control ($n = 2$) or anti-TIMP1 neutralizing antibody ($n = 2$). Scale bar = 200 μm. Arrows indicate positively stained collagen (in blue). Violin plots show data distribution with median (solid line) and quartiles (dashed line) indicated. Statistical analysis was conducted with an unpaired t-test with Welch's correction. **J** Flow cytometry quantification of mammary-resident CD11b^hiLy6G+ neutrophils from NP UTI-bearing mice after 6 doses of IgG control ($n = 7$) or anti-TIMP1 neutralizing antibody ($n = 7$). Error bars represent mean ± S.E.M. Statistical analysis was conducted with an unpaired t-test with Welch's correction; *$p = 0.0314$. **K, L** Masson's trichrome staining of mammary tissue sections and (**L**) quantification of positively stained collagen (in blue) from post-lactation involution (PLI) UTI-bearing mice after 6 doses of IgG control ($n = 4$) or anti-TIMP1 neutralizing antibody ($n = 2$). Scale bar = 200 μm. Arrows indicate positively stained collagen (in blue). Violin plots show data distribution with median (solid line) and quartiles (dashed line) indicated. Statistical analysis was conducted with an unpaired t-test with Welch's correction; *$p = 0.0164$. Source data and p-values are provided as a Source Data file.

(Fig. 4E)[12,42]. We also found elevated levels of Tissue inhibitor of matrix metalloproteinases 1 (TIMP1, 3-fold), a factor previously shown to regulate neutrophilia and collagen remodeling, in the plasma of UTI-bearing mice (Fig. 4E)[12,42]. To confirm that the systemic host response to unresolved UTI is sufficient to drive granulocyte proliferation in a reductionist setting, we cultured total bone marrow cells from healthy mice with plasma from UTI-bearing mice. We found that culture supplementation with plasma from UTI-bearing mice induced a 41% expansion of granulocytes, in comparison to cultures treated with plasma from PBS mice (Supp. Fig. 6H–J). This effect was exclusive to granulocytic cells, given that the abundance of monocytes remained unchanged across the culturing conditions (Supp. Fig. 6J). These findings support the notion that systemic host responses to UTI is accompanied by the increase in many granulocyte-stimulating factors in the circulating plasma, which induce granulocyte expansion in vivo and in vitro.

We next set out to define whether the UTI-induced factor and canonical driver of neutrophil expansion, G-CSF (also known as CSF3), contributed to the expansion of neutrophils in the mammary gland. In doing so, we utilized a previously established protocol, to treat UTI-bearing mice with either control IgG or CSF3 neutralizing antibodies (referred to hereafter as anti-CSF3) by intraperitoneal injection for 2 weeks starting 48 h after UTI establishment (Supp. Fig. 7A)[43]. This treatment did not change the severity of UTI infection, given that both IgG and anti-CSF3 treated UTI-bearing mice continued to display high levels of bacteria in the urine, and histological signs of unresolved infection (Supp. Fig. 7B, C). Evaluation of blood and mammary tissue indicated that anti-CSF3 treatment did not impact the abundance of mammary or circulating CD11b[hi] Ly6G+ neutrophils, suggesting that during an ongoing UTI, inhibition of CSF3 alone is not sufficient to resolve elevated levels of neutrophils (Fig. 4F, G, Supp. Fig. 1F, Supp. Fig. 6B, Supp. Fig. 7D–F). Intriguingly, we found a 58% reduction of collagen abundance in the mammary gland of UTI-bearing mice treated with anti-CSF3, relative to IgG treated animals, indicating a link between neutrophil-inducing factors and collagen accumulation (Fig. 4H, I). In fact, anti-CSF3 treatment has been previously demonstrated to alleviate collagen-induced arthritis, and to control the differentiation and proliferation of fibroblasts, major collagen producers in the mammary gland and more abundant in the tissue of UTI-bearing mice (Fig. 2)[44]. Overall, our findings suggest that multiple UTI-associated factors may play a role in controlling collagen production in mammary tissue.

### Neutralization of TIMP1 levels in UTI-bearing mice restores mammary homeostasis in nulliparous and post-lactation involuting mice

While CSF3 neutralization did reduce collagen accumulation in nulliparous UTI-bearing mice, it did not change mammary neutrophil recruitment, indicating a bi-modal role on immune and collagen-producing cell types during an ongoing UTI (Fig. 4H, I). Interestingly, a more classical collagen remodeling factor, TIMP1, was also more abundant in the plasma of UTI-bearing mice (Supp. Fig. 8A, B). Since TIMP1 is a known regulator of mammary collagen accumulation, involution progression, and adipocyte content, all relevant to phenotypes observed in UTI-bearing animals, we next tested whether neutralization of TIMP1 levels would restore UTI-induced mammary alterations[13,45–48].

In doing so, we utilized previously published protocols, and treated nulliparous UTI-bearing mice with either control IgG or TIMP1 neutralizing antibodies (referred to hereafter as anti-TIMP1) (Supp. Fig. 8C)[49]. TIMP1 neutralization did not alter the severity of UTI, as demonstrated by similar levels of bacteriuria, histopathological abnormalities in the bladders and kidneys of UTI-bearing animals (Supp. Fig. 8D, E). Treatment with anti-TIMP1 decreased the levels of

plasma TIMP1 by 10-fold in UTI mice, in addition to lowering the levels of G-CSF (CSF3, 14-fold) as compared to the IgG control, confirming a co-regulation of circulating factors that respond to UTI (Supp. Fig. 8F).

Histological analyses of anti-TIMP1 treated mice indicated normalization of mammary tissue phenotypes, with a 74% reduction in duct size, a 15% decrease in the abundance of Perilipin+ adipocytes, and a 39% decrease in the amount of collagen content, compared to IgG-treated mice (Fig. 5A–F). Further characterization of the collagen content of IgG control and anti-TIMP1 treated, UTI-bearing mice with confocal SHG confirmed the reduction in collagen deposition upon anti-TIMP1 treatment, and indicated that TIMP1 neutralization was sufficient to prevent altered collagen alignment observed in the mammary tissue from UTI-bearing mice (Fig.5G–I). Alongside resolving UTI-driven, mammary-specific ECM remodeling, TIMP1 neutralization also decreased the abundance of mammary infiltrating CD11b[hi]Ly6G+ neutrophils by 53% compared to control IgG-treated UTI-bearing mice, in contrast to other circulating and mammary infiltrating immune cells populations, which remained unchanged (Fig. 5J, Supp. Fig. 6B, Supp. Fig. 8G). There was no significant difference in the levels of estrogen between the control IgG and anti-TIMP1-treated UTI mice, further supporting that TIMP1, and no other circulating factors that can impact neutrophil homing, drove the alterations to the mammary tissue in UTI-bearing mice (Supp. Fig. 8H)[50,51].

We also evaluated the importance of TIMP1 in coordinating mammary tissue changes by applying the same antibody neutralization regimen in UTI-bearing PLI mice (Supp. Fig. 9A). TIMP1 neutralization did not alter the severity of UTI, as demonstrated by similar levels of bacteriuria, histopathological abnormalities in the bladders and kidneys between UTI-bearing mice in both treatment groups (Supp. Fig. 9B, C). Treatment with anti-TIMP1 antibody decreased the levels of plasma TIMP1 by 6.1-fold in UTI-bearing, in addition to lowering the levels of G-CSF by 2.8 fold as compared to the IgG control (Supp. Fig. 9D).

Treatment with anti-TIMP1 reduced mammary collagen deposition by 63% in UTI-bearing PLI mice, relative to IgG controls, as similarly observed in nulliparous UTI-bearing mice (Fig. 5K, L). However, TIMP1 neutralization did not ameliorate the delayed involution phenotype in UTI-bearing PLI mice, based on similar levels of Cytokeratin 5+ signal (duct content) and β-casein stating (milk residue), between anti-TIMP1 and IgG treated animals (Supp. Fig. 9E, G). Additionally, TIMP1 neutralization did not reduce the number of adipocytes in the mammary tissue of UTI-bearing PLI mice (Supp. Fig. 9G–I), nor affected the abundance of circulating or mammary infiltrated neutrophils/monocytes, given their unchanged levels relative to the mammary tissue of IgG controls (Supp. Fig. 1F, Supp. Fig. 6B, Supp. Fig. 9J, M).

Collectively, our findings elucidate a role for TIMP1 as a critical factor, mediating the influence of the systemic UTI-host-response on mammary tissue health. Our study also provides evidence for the unappreciated role of host responses to a localized infection on regulating cellular and tissue homeostasis of a distal organ, such as the mammary gland.

## Discussion

Using models of induced and treatable infection, we were able to recapitulate clinically relevant UTI etiology and pathobiology in mice. With this model, we demonstrated that the UTI's influence on the mammary tissue was independent of bacterium translocation from the bladder to the mammary tissue, but rather acting through induced systemic effectors. This observation alone suggests against the possibility that our analysis investigated a mastitis-associated phenotype, an infection of breast tissue commonly observed in post-partum women, which is associated with *E.coli* mammary colonization and local tissue inflammatory signals[8,52]. It is worth noting that our study did not address changes to the mammary local microbiota that may occur in response to an ongoing UTI, which

could also play a role in impacting tissue metabolism, development, and immunity[53,54]. Nonetheless, inhibition of UTI-induced circulatory factors, specifically TIMP1, restored mammary tissue health, thus suggesting a master regulator on mammary dynamics in response to the onset of UTI.

Bacterial antigens and bacteria-induced tissue damage triggers chemotactic signals that activate and recruit neutrophils to the infected tissue[36]. Neutrophil stimulating cytokines like G-CSF and CXCL1 are induced in the bladder tissue within 6 hours of a UTI, and our results demonstrated that such levels remain elevated 2 weeks post UTI infection[36,43,55]. While our transcription analysis indicated enrichment of neutrophil recruitment signals on MECs from UTI-bearing mice, a broader analysis demonstrated an expanded neutrophil homing across many other tissues, thus ruling out that neutrophil infiltration during ongoing UTI is mammary specific. Interestingly, targeting of G-CSF, a master regulator of neutrophil release from the bone marrow and tissue homing, with neutralizing antibodies did not impact overall neutrophil levels, but instead substantially decreased the accumulation of collagen in mammary tissue[56]. In fact, G-CSF has been associated with increased expression of collagen genes, thus supporting its multi-functional role regulating mammary phenotypes during an ongoing UTI[57]. Moreover, and given G-CSF also plays a role on regulating the proliferation and differentiation of fibroblasts, these observations support our findings that specific, collagen-producing fibroblast are expanded in nulliparous UTI-bearing mice, further indicating a possible feedback loop of signals that control neutrophil trafficking and collagen remodeling in the mammary gland.

Our findings also elucidated that the MECs responded to the UTI-induced factors in a mammary developmental context-dependent manner. While mechano-sensing and neutrophil recruitment transcriptional programs were altered in major MEC lineages across mammary developmental stages, UTI during post-lactational involution associated with LASP populations expressing signals of both lactation and involution process, a conclusion also supported by tissue analysis. Such signals of delayed involution may be influenced by combinatorial systemic responses to UTI, changes to mammary microenvironment, and overall cellular states, thus illustrating that infections that women are at a higher risk to develop during and after pregnancy could impact the tightly controlled process of tissue reconstruction post-lactation. Importantly, disruption of post-pregnancy/lactation tissue remodeling may provide a niche for oncogenic initiation and malignant outgrowth, and therefore representing a possible alteration that contributes to the incidence of post-partum breast cancer[58,59].

The UTI-elicited alteration of the mammary ECM was reflected in the excessive accumulation of collagen within the mammary fat-pad, along with the presence of increasingly aligned collagen fibers. Increased collagen content within the mammary gland has been described to enhance the elasticity of tissues, resulting in increased stiffness, which subsequently impacts the proliferation, differentiation and lineage commitment state of the neighboring epithelial cells, through stimulation of pro-survival, mechano-sensing pathways[17]. Therefore, the activation of such transcriptional programs, as shown in our scRNA-seq analysis, along with the collagen accumulation affecting tissue stiffness, are features that could create a pro-tumorigenic microenvironment and generate windows of opportunity for the development and progression of malignant cells, across developmental stages of the gland[17,60].

But most importantly, our study elucidated the link between TIMP1 levels, mammary tissue alterations, and an ongoing UTI. Increased TIMP1 levels has been associated with granulopoiesis, neutrophilia, and collagen accumulation and altered adipogenesis in several tissues, including but not limited to the mammary gland, cardiac tissue, lungs and kidneys[12,17,42,61–63]. In the mammary gland,

prolonged expression of TIMP1 has been shown to impair post-lactation regression of ductal structures, and persistent milk production, therefore matching the phenotype observed in UTI-bearing mice during post-lactation involution[45,48]. Here, utilizing a neutralizing antibody approach targeting TIMP1, we were able to fully rescue the abnormal neutrophil infiltration, adipocyte expansion, ductal abnormality, and collagen deposition in the mammary tissue of UTI-bearing mice. TIMP1 neutralization restored homeostasis in the mammary gland in UTI-bearing mice without affecting disease presentation in the bladder or kidney or impacting circulating estrogen or corticosteroids levels, supporting that its mammary specific impact is central to the influence of UTI on mammary health.

But where does TIMP1 come from in UTI-bearing mice? TIMP1 has predominantly been detected in polymorphonuclear cells and resident tubular cells in the kidneys, suggesting that both polymorphonuclear cells, like neutrophils, and tissue-derived TIMP1 could contribute to the increased pool of TIMP1 in circulation in UTI-bearing mice[64]. While our analyses did not differentiate between TIMP1 proteins bound to the neutrophil surface or actively secreted by other cell subtypes, the elevation of TIMP1 levels in the plasma of UTI-bearing mice supports that it is secreted into circulation[65].

Nonetheless, our findings illustrate a paradigm in mammary biology and development as they show that the plasticity of the mammary gland, canonically thought to be largely hormone-responsive, can also be modulated by other common health events, such as infection. The changes we observe in the mammary gland during an ongoing UTI bear relevance to breast density and fibrosis, changes that have the potential to foreshadow increased risk of future malignancy. This raises the hypothesis that perhaps other systemic events in a woman's life, outside of female hormones and age, should be considered with respect to how they impact breast health. This may be of special concern for women at risk for developing breast cancer, such as those bearing germline mutations that predispose to breast cancer development, family history of breast cancer, or other breast-cancer related lifestyle factors (parity, BMI, etc). The further implication of these UTI-induced changes within mammary tissue on breast oncogenesis must be further investigated in mouse models of breast cancer, and evaluated with large scale epidemiological datasets.

## Methods

### Regulatory statement
Our research complies with all relevant ethical regulations in accordance with CSHL Institutional Animal Care and Use Committee guidelines, chaired by Lisa Bianco.

### Animals
Nulliparous and timed-pregnant (gestation day E11-E15), female, C57BL/6 J mice were purchased from The Jackson Laboratory. All animals were housed at the CSHL shared Laboratory Animal Resource under a 12 hour. light/dark cycle, with controlled temperature and humidity at 72°F and 40-60%, respectively, and with access to dry food and water ad libitum, unless otherwise specified. Animals were euthanized via Carbon dioxide ($CO_2$) inhalation. All animal experiments were performed in accordance with CSHL Institutional Animal Care and Use Committee guidelines. All animal related analysis (UTI, offspring weaning, and treatments) were performed as described in Supplementary Methods.

### Tissue collection and processing for histological analyses
Mammary gland (left inguinal), bladder, kidney, spleen, lungs, liver, intestines, and pancreas were harvested at the experimental end-point and immediately fixed in 4% paraformaldehyde in 1X PBS (16% solution, Electron Microscopy Sciences, cat# 15711) at 4°C overnight,

followed by storage in 1X PBS at 4°C, until paraffin embedding. Tissue processing, paraffin embedding, sectioning, staining, digestion and analysis were performed as described in Supplementary Methods.

**RNA sequencing (scRNA-seq) library preparation and analysis**
scRNA-seq libraries were prepared, normalized, integrated, batch corrected and analyzed as described in Supplementary Methods.

**Reporting summary**
Further information on research design is available in the Nature Portfolio Reporting Summary linked to this article.

## Data availability

scRNA-seq were deposited into BioProject database under accession number PRJNA855880. Datasets used on Fig. 2 and Fig. S3 ('no-UTI' -seq) were previously deposited into BioProject database under accession number PRJNA677888. The remaining data are available within the Article, Supplementary Information or Source Data file. Source data are provided with this paper.

## Code availability

This manuscript does not report original code. Code and featured feature matrixes are available at https://github.com/dosSantosLabCSHL/scRNA-NP-P-UTI-SC-2022. Any additional information required to reanalyze the data reported in this paper will be made available upon request.

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

## Acknowledgements

We thank the CSHL Cancer Center Shared Resources (Laboratory Animal Resource, Histology, Microscopy, Flow Cytometry and Single Cell Sequencing core facilities) for their assistance in the completion of the described study and the entire dos Santos lab for their insights and helpful discussions related to this manuscript. This work was performed with support by the CSHL and Simons Foundation (C.O.D.S. and D.A.T.), the Lee MacCormick Edwards Charitable Foundation (C.O.D.S.), the CSHL and Northwell health affiliation (C.O.D.S.), the Pershing Square Sohn Prize for Cancer Research (C.O.D.S.), the Robertson foundation (C.O.D.S.), the Breast Cancer Research Foundation (C.O.D.S.), the NIH/NCI grant R01CA248158-01 (C.O.D.S.), NIH/NIA grant R01 AG069727-01 (C.O.D.S.), the NIH/NCI 1R01CA284630 (C.O.D.S.), the NIH NCI grant F30CA281082 (S.M.L.), the NIHGM T32GM008444 (S.M.L.), and the NIH NCI grant 1R01CA2374135 (M. E.). CSHL Cancer Center Shared Resources are supported by the CSHL Cancer Center Support Grant 5P30CA045508.

## Author contributions

Conceptualization: C.O.D.S.; Methodology: S.H., S.M.L., S.L.C., M.K.C., D.C., A.V.H.S., G.J., X.H., G.C., M.F.C., I.A.D., A.A.B., E.H., T.H., D.T. and M.E.; Investigation: S.H., S.M.L., S.L.C., M.K.C., G.J., E.H., J.E.W., Visualization: S.H., S.M.L., S.L.C., M.K.C., X.H. Funding acquisition: C.O.D.S.; Project administration: C.O.D.S.; Supervision: C.O.D.S.; Writing – original draft: C.O.D.S, S.L.C., S.H. and S.M.L. Writing – revised draft: C.O.D.S, S.H. and S.M.L.

## Competing interests

The authors declare no competing interests.

## Additional information

[1]Cold Spring Harbor Laboratory, Cold Spring Harbor, NY, USA. [2]Stony Brook University, Graduate Program in Genetics, Stony Brook, NY, USA. [3]CSHL School of Biological Sciences, Cold Spring Harbor, NY, USA. [4]Department of Cell Biology and Physiology. School of Medicine in St. Louis. Washington University, St. Louis, MO, USA. [5]SUNY Downstate Health Sciences University, Neural and Behavior Science, Brooklyn, NY, USA. [6]Department of Comparative Medicine, University of Washington, Seattle, WA, USA. [7]Department of Cell Biology, Department of Oncology, School of Medicine, Johns Hopkins University, Baltimore, MD, USA. [8]These authors contributed equally: Samantha Henry, Steven Macauley Lewis, Samantha Leeanne Cyrill. ✉e-mail: dossanto@cshl.edu

