## [Peer Review File · Nature Communications]

Host response during unresolved urinary tract infection alters female mammary tissue homeostasis through collagen deposition and TIMP1REVIEWER COMMENTS

Reviewer #1 (Remarks to the Author): with expertise in mammary genome, mammary glands and pregnancy

This is a thought-provoking study showing UTI induced perturbations in mammary tissue, including ductal hyperplasia, altered immune composition, and abnormal collagen accumulation. The authors also show delayed mammary regression / involution as demonstrated by increased milk gene expression as well as an expansion of neutrophils. Lastly, they propose that TIMP-1 is a master regulator of mammary homeostasis. This conclusion is based on the observation that TIMP-1 levels are elevated in their model and suppression of TIMP-1 reverts the pathological phenotype, at least in part.

Critique

Inoculation of the bladder with a uropathogenic E coli strain serves as experimental paradigm. This reviewer wonders if the pathological effects observed in this study are caused by increased corticosteroid levels? Have the authors measured these? There is plenty of evidence that glucocorticoids drive mammary development and differentiation.

The authors also investigated the process of forced involution and concluded that mammary epithelium in UTI mice retained some differentiation. This conclusion is based on elevated expression levels of beta casein (Csn2) (Fig 2C-F). Csn2 is a sub optimal marker of epithelial cell differentiation as it is also expressed in some immune cells. It would have been helpful to see the raw gene expression data of Csn2 and other more reliable milk protein genes, such as Csn1s2b and Wap.

The quality of scRNA-seq depends on cell viability and cell differentiation status. Isolation of single cells from mammary tissue involves collagenase treatment that certainly impacts cell behavior, especially after 90 minutes. I wonder whether the authors have conducted bulk RNA-seq from fresh tissue to validate some of their scRNA-seq results. It should be possible to deconvolute bulk RNA-seq data and identify cell types and differentiation status.

The authors propose TIMP-1 as a master regulator of mammary homeostasis. Several

mouse studies have explored the role of TIMP-1, both upon deletion of the gene and ectopic overexpression. The authors need to explore this literature and determine to what extent their findings agree or disagree with current knowledge. Maybe the authors should investigate TIMP-1 ko mice.

Timp-1 gene ko mice (selection of papers)

<https://pubmed.ncbi.nlm.nih.gov/10652195/>

<https://pubmed.ncbi.nlm.nih.gov/27194531/>

<https://pubmed.ncbi.nlm.nih.gov/11732988/>

<https://pubmed.ncbi.nlm.nih.gov/18560439/>

<https://pubmed.ncbi.nlm.nih.gov/15598866/>

Ectopic TIM-1 expression

<https://pubmed.ncbi.nlm.nih.gov/10395785/>

Reviewer #2 (Remarks to the Author): with expertise in neutrophils and urinary infections

In this manuscript, the authors investigated UTI-induced changes in the mammary gland. They describe ductal hyperplasia, altered immune composition, and abnormal collagen accumulation. Mammary epithelial cells upregulated genes for mechano-sensing and immune recruitment pathways. In glands undergoing post-partum involution, the presence of a UTI delays mammary regression, marked by the presence of residual milk protein and unregressed ductal structures. UTI also induced expansion of circulating and mammary-infiltrating neutrophils, and neutralization of the UTI-induced circulating factor TIMP1 restored normal mammary tissue homeostasis.

The findings are clearly presented and add important information how an infected organ, here the urinary bladder, affects distal mammary gland inflammation and remodeling.

Major concern:

Why was the statistics be performed ROI- and not sample-based? In case e.g. 50 ducts per gland were analyzed, one mean or median should be used as „n”, and not a value of “50”.

This concern refers to several histology-based figures. This might significantly change most of the statements in the manuscript.

Minor concerns:

Fig. S1B-E: Surprising that even 2 weeks after infection, the frequency of neutrophils is increased. Pls consider to reanalyze the data.

scRNAseq. Figure 1F: You might study Batch effects here, since data were not acquired with the same instrument and settings.

Please introduce the abbreviation NEC better in the text, as in the Figure 1F itself => Nulliparous Epithelial Cell Clusters, and also in Figures S3. Otherwise the meaning of NEC was not directly clear to me.

Fig. 3A and B: Please elaborate (e.g. data in Supplement) on the genes associated with this neutrophil recruitment signature

Fig. 3D to E: Comment: CD11b lo Neutrophils seems more frequent than CD11b hi neutrophils. This seems rather odd, but a rather non-activated CD11b lo phenotype might fit to the data in Figure 3D.

The Figure M1 is not mentioned in the text, but important to understand how the gating was performed. Please mention the figures accordingly.

Please provide staining controls (FMO) for Figure M1 "Live cells" and for the CD11b versus Ly6G staining. FMO controls should be included for both Ly6G and CD11b.

Figure 3F: Please also provide lower magnified images in addition to the included zoom images. Please omit shown two images of the same condition.

Mouse infections and TMS treatment: When were the mice treated with TMS. I cannot find information in the figures or M&M section.

Figure 3C: Please indicated the time-point of analysis in the legend. This is particularly interesting as the number of CD11b lo seems extremely high among all leukocytes, and since Figure S6 indicates strongly less neutrophil frequencies at day 3 post-infection.

Figure S1C to E: Please refer to the details here: i.e. gating strategies, Marker panels in Fig. M1.

Figure 1G: Please elaborate on how the communication signature was identified. Which genes contribute to the communication?

Discussion

„Using models of an induced UPEC infection, we were able to recapitulate clinically relevant UTI etiology and pathobiology in mice, while demonstrating that the UTI’s influence on the mammary tissue was independent of the causative bacterium.“ The statement that „...that the UTI’s influence on the mammary tissue was independent of the causative bacterium.“ is incorrect. The influence on the mammary tissue is dependent on the causative bacterium. I believe that the authors wanted to state that bacteria in the mammary gland are not causative for the local changes. Please revise and clarify.

„Neutrophils are minimally present in mammary tissue of healthy nulliparous mice, indicating they are part of a much larger immune program that plays a role in immune surveillance in the gland (42, 43).“ This conclusion is unclear to me. Why does low numbers of neutrophils in the mammary gland indicate “..part of a much larger immune program..”. This conclusion seems too far-fetched.

“...associated with increased susceptibility and severe inflammatory responses to UTIs (47) The...”. Point missing.

“Increased collagen content within the mammary gland has been described to have exponentially higher viscoelasticity, ...”. Is this association correct? Increased collagen is usually associated with less elasticity.

„Interestingly, studies analyzing extended antibiotic regimens for persistent UTIs did not show differences in the general incidence of breast cancer, suggesting that untreatable and persistent cases of UTI may correlate with an increased risk for cancer development (40).“ Is disagree with this conclusion. Please clarify why no changes in cancer incidence suggest increased risk for cancer.

Reviewer #3 (Remarks to the Author): with expertise in mammary gland, single cell genomics

This paper investigates how a urinary tract infections affect distal organs, focusing on tissue alterations in the mammary gland. The work utilises single cell RNA sequencing data to understand the transcriptional changes in mammary epithelial cells and reports upregulation of machano-sensing and immune recruitment pathways. Using pregnancy models, the paper also reports delayed involution in the presence of UTIs. This is an interesting study and topic but requires further work to address outstanding questions as described in the comments below.

Major Points

Figure 1E: Summary graphs 1B and 1D are given to summarise the overall data from multiple glands and fields of view for ductal lumen area and collagen, respectively. Please could a similar summary graph be made to represent all of the images and data gathered from picosirius red staining in addition to the single representative screenshots shown?

Figure 1F & Supplemental Figure 3:

- NEC cluster naming is not very informative and difficult to follow, we recommend using more informative population names
- UMAP shows 3 progenitor populations which cluster far away from each other. With the exception of some evidence of a small population of long-lived and primarily quiescent stem cells, many tracing studies now indicate that in the stable in-vivo adult mammary gland, basal and luminal lineages are self-maintained by unipotent progenitors. As mice are 7-9 weeks old, it would be unexpected to see bipotent progenitors, and more expected to see unipotent basal progenitors and ER-/PR- and ER+/PR+ luminal progenitors. Please could the authors review this data.

- Supplemental figure 3a uses many non-canonical gene markers. Please could the authors provide references and/or justification for why these selected non-canonical gene markers are indicative of the cell type they are used to represent.
- Additionally, it appears some canonical cell type markers are assigned to incorrect cell populations. For example, immune cell markers CF52 and CD74 are categorised as bipotent progenitor markers. Please could the authors provide justification for this or update these as appropriate.
- The YAP signature used is from a paper which used YAP overexpression in human mammary cell line MCF10As, or from mouse liver tissues or mouse immortalised fibroblasts (not mammary and also immortalised). Please could the authors provide justification as to why this a valid signature for YAP in the normal mouse mammary gland or update as appropriate.

Naming of populations is not in keeping with previous literature precedent. In particular, luminal ductal population lacks a reference or explanation and is not previously widely described. Please update where possible and appropriate to standard nomenclature.

Generally, in figure S3a some markers were assigned unexpected population names based on previous literature. For example:

- Markers for luminal ductal clusters and luminal progenitor cluster (NEC8) appear more like hormone sensing as they score highly for canonical markers such as ESR1 and AREG.
- Separate NEC9 clustering and higher expression of c-kit and Aldh1a3 suggests these may be hormone sensing progenitors.
- Aldh1a3 was assigned as a luminal alveolar marker but is canonically used as a luminal progenitor marker.

Please could the authors justify why these examples were assigned as stated or update as appropriate?

Please could the authors check marker gene set labels in figure S3a as these do not align with labels in S3b. For example, the population name for NEC8 being ascribed to LPs has no marker gene set shown in S3a?

Major: Figure 2 and Supplemental Figure 4:

- Two obvious variables between NPN vs Pseudo without nursing- can a better comparator be made e.g. Natural pregnancy without nursing or Pseudo plus nursing (adoption)?
- Please could the authors clarify why pseudopregnancy was selected and outline what benefit this offers over the natural pregnancy alone model? Pseudopregnancy is not representative of something that happens often. Furthermore, the data shown (discussed further below) suggests these two pregnancy models vary quite significantly in their phenotype, which means they cannot be compared.
- Pseudo control group without treatment needed to determine if UTI / TMS effect or pseudo pregnancy effect? (S4D suggests some of the UTI group have low bacteria counts. Are these wrongly classified?)
- A control group given TMS alone might help justify that the UTI + treatment group are a valid control group by showing that additional treatment does not affect the mammary gland.
- Evidence that the choice of mouse model might have a significant effect on the mammary gland phenotype being observed is indicated by the lack of correlation of the phenotype with the bacterial titre/degree of infection. For example, figures S4B-S4F shows that bacterial titre in UPEC mice are on average ~2x higher than pseudopregnancy mice. However, cytokeratin 5 (representing ductal structures) are more than double in pseudopregnancy mice relative to UPECs, and additionally, beta-casein retention is higher in pseudopregnancy mice relative to UPECs. The fact that the choice of mouse model significantly affects the phenotype may confound the cause and effect of the UTI, and more interestingly, implies that there may be additional model-specific mechanisms that affect the mammary gland phenotype which the authors could choose to pursue further.

Major :Figure 2J and 2I:

- Please provide pseudo pregnancy PBS control or justification for absence of no baseline defined for pseudo pregnancy. Since natural pregnancy PBS control has similar collagen content to the pseudopregnancy with the UTI, it is unclear if the baseline for the pseudopregnancy vs natural pregnancy is different and it is difficult to determine whether pseudopregnancy + treatment is a suitable control.
- The pseudopregnancy + UTI + treatment is a more complicated control and no control for antibiotic only is shown therefore it cannot be ruled out that TMS has an effect by itself other than curing the infection.

- Could natural and pseudo pregnancy conditions be plotted on the same graph for easier and more robust statistical comparison?

-

Section 3: UTI induces neutrophil recruitment

- Given it already well established that response to infection includes neutrophil recruitment, and given that another study on *Helicobacter Hepaticus* shows that the mammary gland recruits neutrophils in response to infection with a uropathogenic strain of *E. coli*, some further experiments may be required to make this section novel. For example, looking at neutrophil recruitment in other distal organs to see if this is a mammary specific response, or looking at other types of distal infection to see if this also has mammary specific neutrophil recruitment.

- The text references a 93% increase in Cd11bhi neutrophils in PBS and TMS groups. Was this an exact 93% increase Cd11bhi neutrophils in both PBS and TMS groups? Otherwise, we would request the authors report the PBS and TMS respective values.

- TIMP1 increases circulatory factors which increases neutrophil expansion, however it is currently unclear if TIMP1 induces changes directly or indirectly via the circulatory factors. Please could the authors clarify if and how additional circulatory factors were ruled out as causes of the increase in mammary resident neutrophils and alteration to the ECM and ductal structures. If not already investigated, further additional work to rule out circulating factors as a cause would be insightful.

- We recommend the same in-depth analysis pre- and post- anti-TIMP1 administration to understand if treatment reverses all the phenotypes previously reported in the paper observed on infection. If there is not a complete return to the original phenotype, it might be of interest to explore the consequences or causes of that further.

General Comments

- Frequent switching and comparison between natural and pseudo pregnancy is quite difficult to follow. The acronyms used are not easy to follow. Please review whether more self explanatory acronyms could be used.

- Figure legends could be much more informative and should define the acronyms used in each legend to more easily stand alone from text.

- Discussion is slightly speculative (Lines 315-318) and in places difficult to follow (Lines 331-334). Please make more focused and data driven.

Technical Comments

- Different routes of administration: Bladder inoculation vs transurethral administration.

Can a justification be provided for this?

- Please could the authors provide details for recombinant TIMP1 protein (rTIMP1) along with details on how the dose concentration & treatment duration were selected?

Concentration used was much higher than concentration measured.

Figure 1F & Supplemental Figure 3:

- Data suggests potential doublet populations within the data set. Please could the authors provide justification for why stromal cells were excluded using MACS beads and not FACS? Additionally, please could they provide details of any steps (e.g. flow cytometry) performed to validate this as an effective method for excluding lineage positive cells?
- The inclusion of flow cytometry data generated from samples post MACS bead filtering would be beneficial to demonstrate this as an effective approach to exclude lineage positive cells as would the inclusion of doublet scoring and exclusion in scRNA-seq QC pipeline.
- Minor: Data processing and analysis for scRNA-seq: Please provide information on the type of batch correction or normalisation used along with a justification for why this was appropriate.

Minor: Figure S5:

- Please provide information and justification on why PEC was batch corrected, normalised and clustered separately to the NECs?
- Please update PEC# to more informative population names.
- Please provide justification as to why all of the UTI groups in this figure are pseudopregnancy mice and why these were used instead of natural pregnancy group or a combination of the two.

Minor: Figure S5C:

- We recommend using gene signatures composed of multiple genes to represent pathways rather than single genes selected to represent whole pathways as this is likely to be more representative.
- We recommend plotting the average expression of individual mice rather than plotting

individual cells and repeating P-values with individual mouse average expression.

(Statistically comparing the difference in expression of single cells assumes each cell can be considered as completely independent events. However, bias is introduced by cells coming from the same mice and being prepared at the same time.

- Y axis labelling- requires clarification as to units.

Responses to Reviewer Comments

We would like to thank the reviewers for taking the time to review our manuscript and for their insightful and helpful comments. We have revised our manuscript to address their concerns to the best of our ability and are confident that the changes made have strengthened our findings.

Please find below our responses (in blue font) to each of the reviewers' concerns.

Reviewer #1 (Remarks to the Author): with expertise in mammary genome, mammary glands and pregnancy

This is a thought-provoking study showing UTI induced perturbations in mammary tissue, including ductal hyperplasia, altered immune composition, and abnormal collagen accumulation. The authors also show delayed mammary regression / involution as demonstrated by increased milk gene expression as well as an expansion of neutrophils. Lastly, they propose that TIMP-1 is a master regulator of mammary homeostasis. This conclusion is based on the observation that TIMP-1 levels are elevated in their model and suppression of TIMP reverts the pathological phenotype, at least in part.

Point 1: Inoculation of the bladder with a uropathogenic E coli strain serves as experimental paradigm. This reviewer wonders if the pathological effects observed in this study are caused by increased corticosteroid levels? Have the authors measured these? There is plenty of evidence that glucocorticoids drive mammary development and differentiation.

Answer to Point 1: To address the reviewer's concern, we quantified corticosterone levels in plasma of nulliparous mice from PBS-control, UTI-bearing, and UTI-bearing + antibiotic TMS treatment experimental groups. We found no difference in plasma corticosterone between the three conditions, suggesting that the observed phenotypes in UTI-bearing mice were not driven by altered corticosterone levels (**Fig. S2D**).

Point 2: The authors also investigated the process of forced involution and concluded that mammary epithelium in UTI mice retained some differentiation. This conclusion is based on elevated expression levels of beta casein (Csn2) (Fig 2C-F). Csn2 is a sub optimal marker of epithelial cell differentiation as it is also expressed in some immune cells. It would have been helpful to see the raw gene expression data of Csn2 and other more reliable milk protein genes, such as Csn1s2b and Wap.

Answer to Point 2: Our analysis was focused on β -casein inside ductal structures, rather than in isolated cells, thus capturing overall milk protein accumulation, rather than β -casein protein levels in isolated cells. To further address the reviewer's concern about the expression of Csn2 on non-epithelial cells, we captured immunofluorescence images with of mammary tissue from UTI-bearing during post-partum involution, indicating β -casein staining exclusively in the lumen of ductal structures, and no detectable staining in the lymph node (immune cell hub) (**Fig. R1-A**).

A second concern raised by the reviewer suggests that “*It would have been helpful to see the raw gene expression data of Csn2 and other more reliable milk protein genes, such as Csn1s2b and Wap*”. Csn2, Csn12b, and Wap mRNA levels have been shown to be downregulated early during involution (D4) (Stein et al. 2004), and therefore their levels are expected to be downregulated in our model (~2 weeks post end of lactation) independently of UTI. Moreover, milk protein accumulation inside of mammary duct structures (milk stasis) has been previously shown to indicate mammary delayed involution states, an observation that we investigated by quantifying the levels of β -casein protein. However, to address the reviewer’s critique, we performed an unsupervised gene expression analysis using GSEA, and identified that luminal alveolar cells from post-lactating UTI-bearing mice have decreased enrichment of pathways linked with involution progression (**Fig. 3H**), thus supporting our overall conclusion.

Point 3: The quality of scRNA-seq depends on cell viability and cell differentiation status. Isolation of single cells from mammary tissue involves collagenase treatment that certainly impacts cell behavior, especially after 90 minutes. I wonder whether the authors have conducted bulk RNA-seq from fresh tissue to validate some of their scRNA-seq results. It should be possible to deconvolute bulk RNA-seq data and identify cell types and differentiation status.

Answer to Point 3: While we acknowledge that tissue dissociation protocols may have limitations, multiple steps were taken to ensure our scRNA-seq libraries were of good quality. To address the reviewer’s comments, we have expanded the methods description to include critical information regarding sample quality and dataset QC: *Cell viability was ascertained, using Trypan blue staining, prior to preparation of scRNA-seq libraries and only samples with a viability >80% were sequenced. Samples with less than 60% viability were not utilized for the generation of scRNA-seq libraries. Computational quality control and filtering were used to exclude low quality cells in the scRNA-seq datasets, excluding those with uncharacteristically low or high feature counts and read output and, high mitochondrial content (Fig. R1-B).*

It is worth to mention that all methods adopted for mammary epithelial cell processing, either tissue snap freezing, mechanic/enzymatic-based, or fluorescence-assisted cell sorting of populations of interest, can influence the quality and information provided by bulk RNAseq datasets. Therefore, even when using bulk sequencing approaches, one has to ensure data quality with robust quality checks.

Point 4: The authors propose TIMP-1 as a master regulator of mammary homeostasis. Several mouse studies have explored the role of TIMP-1, both upon deletion of the gene and ectopic overexpression. The authors need to explore this literature and determine to what extent their findings agree or disagree with current knowledge. Maybe the authors should investigate TIMP-1 ko mice.

Timp-1 gene ko mice (selection of papers)

<https://pubmed.ncbi.nlm.nih.gov/10652195/>

<https://pubmed.ncbi.nlm.nih.gov/27194531/>

<https://pubmed.ncbi.nlm.nih.gov/11732988/>

<https://pubmed.ncbi.nlm.nih.gov/18560439/>

<https://pubmed.ncbi.nlm.nih.gov/15598866/>

Ectopic TIM-1 expression

<https://pubmed.ncbi.nlm.nih.gov/10395785/>

Answer to Point 3: We agree that transgenic knockout or overexpression rodent models have been valuable in understanding the function of endogenous TIMP1 in the mammary gland and other organs. However, genetic ablation of TIMP1 would drive inherent alterations to the mammary gland, even before the onset of UTI, that would require comprehensive characterization, which is beyond the scope of this study. Instead, our approach to utilize TIMP1 neutralizing antibodies, over a transgenic knockout mouse model, allowed to reserve a phenotype that was induced by UTI, and based on TIMP1 levels, within a wildtype genetic background.

To address the reviewer's concern, and given that transgenic TIMP1 overexpression has been documented to influence adipogenesis in various tissues, including the mammary gland (Alexander *et al.* 2001), we have added an analysis of mammary adipocytes in UTI-bearing mice to the revised manuscript. Briefly, we found that nulliparous UTI-bearing mice had a higher number of Perilipin+ adipocytes than control PBS mice and, treatment of nulliparous UTI-bearing

mice with a TIMP1 neutralizing antibody ablated this phenotype. These results support that increased TIMP1 levels, in response to UTI, does influence mammary tissue homeostasis, as observed with TIMP1 overexpression (Fig.1D-E and Fig.5C-D).

Reviewer #2 (Remarks to the Author): with expertise in neutrophils and urinary infections

In this manuscript, the authors investigated UTI-induced changes in the mammary gland. They describe ductal hyperplasia, altered immune composition, and abnormal collagen accumulation. Mammary epithelial cells upregulated genes for mechano-sensing and immune recruitment pathways. In glands undergoing post-partum involution, the presence of a UTI delays mammary regression, marked by the presence of residual milk protein and unregressed ductal structures. UTI also induced expansion of circulating and mammary-infiltrating neutrophils, and neutralization of the UTI-induced circulating factor TIMP1 restored normal mammary tissue homeostasis. The findings are clearly presented and add important information how an infected organ, here the urinary bladder, affects distal mammary gland inflammation and remodeling.

Major concern:

Point 1: Why was the statistics be performed ROI- and not sample-based? In case e.g. 50 ducts per gland were analyzed, one mean or median should be used as „n”, and not a value of “50”. This concern refers to several histology-based figures. This might significantly change most of the statements in the manuscript.

Answer to Point 1: The mammary gland has considerable heterogeneity in tissue size and composition, including the distribution of collagen deposits. Therefore, for the imaging analyses in this study we employed a hierarchical/spatial repeated measures approach commonly used in mammary histological assessment (O'Brien *et al.* 2010, Lyons *et al.* 2011, Maller *et al.* 2013, Guo *et al.* 2022). In reporting a singular mean/median, information about observational variability is lost and precision of data reporting is lowered (Lazic 2021).

In order to fully address the reviewer's concern, all of datasets were re-analyzed by the CSHL biostatistician, and now co-author in this manuscript, Dr. Taehoon Ha, Briefly, datasets were assessed for normality and log transformed, to adjust to a normal distribution. Statistical significance testing was carried out using a linear mixed effects model to compare conditions while accounting for both, fixed effects (group) and random effects (data points nested within the same biological replicate).

A detailed description of the biostatistical analyses has been included in the supplementary methods. Additionally, all bar & scatter plots for image-based analyses have been replaced with violin plots to better reflect the data distribution and p-values have been amended, as described above, in the revised manuscript.

Minor concerns:

Point 2: Fig. S1B-E: Surprising that even 2 weeks after infection, the frequency of neutrophils is increased. Pls consider to reanalyze the data.

Answer to Point 2: As suggested by the reviewer, we have revisited and revised all of the datasets, staining's, and experimental time points, and we confirmed that all of the analysis was indeed performed at 2-weeks post infection.

At the experimental end-point (2-weeks post infection), all UTI-bearing mice analyzed had a subsided, but still active UTI. This was supported by histopathological assessment of the bladder and kidney tissues that confirmed the presence of UTI-induced alterations (assessed by Dr. Erby Wilkinson, pathology at CSHL Cancer Center and co-author in this manuscript) (**Fig. S1D**). Analysis of plasma cytokines of UTI-bearing mice (nulliparous and post-lactation involution), at 2 weeks post-infection, revealed presence of immune chemo-stimulants, like G-CSF and CXCL1, that classically elicit a granulocytic response (**Fig. 4E** and **Fig. S7A**). Furthermore, we confirmed increased neutrophil levels with several methods: recruitment signature from MECs harvested from UTI-bearing mice at 2 weeks post infection (**Fig.4A**); b) Flow cytometry analysis of CD11b+Ly6G+ mammary resident cells and circulating cells from UTI-bearing mice at 2 weeks post infection (**Fig. S1F-G, Fig.4B-C**); c) immunofluorescence staining (**Fig.4D** and **Fig. S6E-F**). We also provided neutrophil Flow cytometry analysis of mammary tissue 72 hrs post UTI infection, and at this time point, there is no increase on the abundance of neutrophils (**Fig. S6B-C**).

Point 3: scRNAseq. Figure 1F: You might study Batch effects here, since data were not acquired with the same instrument and settings.

Answer to Point 3: All scRNA sequencing datasets were integrated and normalized to control for any batch effects using the FindIntegrationsAnchors() function followed by the IntegrateData() function. To address the reviewer's concerns, we have included an improved description of all data pre-processing and batch effects correction in the supplementary methods.

Point 4: Please introduce the abbreviation NEC better in the text, as in the Figure 1F itself => Nulliparous Epithelial Cell Clusters, and also in Figures S3. Otherwise the meaning of NEC was not directly clear to me.

Answer to Point 4: We have re-labeled all figures to better reflect the experimental conditions.

Point 5: Fig. 3A and B: Please elaborate (e.g. data in Supplement) on the genes associated with this neutrophil recruitment signature

Answer to Point 5: The gene signature was collected from a previous publication that investigate how neutrophils get recruited by MECs (Neeli et. a. J. Immunology, 2008, cited on supplemented information). To address the reviewer's request, we have added supplementary tables 1-6 containing detailed gene lists and expression data (average log 2-fold change and adjusted p-values) for all the gene signatures used in this study.

Point 6: Fig. 3D to E: Comment: CD11b lo Neutrophils seems more frequent than CD11b hi neutrophils. This seems rather odd, but a rather non-activated CD11b lo phenotype might fit to the data in Figure 3D.

Answer to Point 6: There was no statistically significant differences observed in the number of non-activated CD11b^{lo} Ly6G⁺ cells between the compared groups. Both CD11b and Myeloperoxidase are known markers for neutrophil activation (Lau et al. 2004), suggesting that the CD11b^{high} Ly6G⁺ cells and Myeloperoxidase⁺ cells, that are elevated in mammary glands from UTI-bearing mice, are a mature neutrophil population.

Point 7: The Figure M1 is not mentioned in the text, but important to understand how the gating was performed. Please mention the figures accordingly. Please provide staining controls (FMO) for Figure M1 "Live cells" and for the CD11b versus Ly6G staining. FMO controls should be included for both Ly6G and CD11b.

Answer to Point 7: In order to address the reviewer's comments, we have included the following information to the methods section, to demonstrate how single staining controls and gating strategies were performed: *To assess spectral overlap between the chosen fluorophores, we used single color cell controls and their fluorescence profiles are shown below (Fig. R2-B).* Additionally, we have included citations for Figure M1 in the figure legends wherever appropriate.

Fig. R2-B. Single fluorophore-stained controls for flow cytometric analysis. Histogram plots for unstained cells (darker histogram) and cell controls stained with individual fluorophore-labeled antibodies (lighter histogram). The plots show fluorescence intensity (x-axis) across all channels and cell count (y-axis). Here, we show that the selected fluorophores had minimal non-specific spill-over and we were able to confidently identify cell populations upon compensation

Point 8: Figure 3F: Please also provide lower magnified images in addition to the included zoom images. Please omit shown two images of the same condition.

Answer to Point 8: The inclusion of two representative images per analysis is to show the degree of variability of the quantified phenotypes, a transparent approach that allows for appropriate data interpretation. All image quantification was performed on whole mammary tissue, thus excluding the need of lower magnified images, which would not be very informative given the size of the tissue.

Point 9: Mouse infections and TMS treatment: When were the mice treated with TMS. I cannot find information in the figures or M&M section.

Answer to Point 9: We have clarified the description of TMS treatment regimen on the supplementary methods, as follows *“EQUISUL-SDT antimicrobial oral suspension (Covetrus, cat# 048889) containing Trimethoprim-Sulfamethoxazole (TMS) was added to MediGel® Sucralose cups (2 oz, ClearH₂O), at a concentration of 0.48 mg/ml. The TMS cups were placed in the animal cages as the sole source of hydration for the duration of the experiment starting 48 hours p.i.. Urine samples from TMS-treated mice were collected and titrated on LB agar plates to assess UTI resolution”.*

Point 10: Figure 3C: Please indicated the time-point of analysis in the legend. This is particularly interesting as the number of CD11b lo seems extremely high among all leukocytes, and since Figure S6 indicates strongly less neutrophil frequencies at day 3 post-infection.

Answer to Point 10: We have updated all figure legends with details about data time-points.

Point 11: Figure S1C to E: Please refer to the details here: i.e. gating strategies, Marker panels in Fig. M1.

Answer to Point 11: We have updated all figure legends pertaining to flow cytometry analyses to cite the methods Figure M1.

Point 12: Figure 1G: Please elaborate on how the communication signature was identified. Which genes contribute to the communication?

Answer to Point 12: We have added supplementary tables 1-6 containing detailed gene lists and statistical data for all the gene expression signatures used in this study.

Point 13: “Using models of an induced UPEC infection, we were able to recapitulate clinically relevant UTI etiology and pathobiology in mice, while demonstrating that the UTI’s influence on the mammary tissue was independent of the causative bacterium.” The statement that „...that the UTI’s influence on the mammary tissue was independent of the causative bacterium.” is incorrect. The influence on the mammary tissue is dependent on the causative bacterium. I believe that the authors wanted to state that bacteria in the mammary gland are not causative for the local changes. Please revise and clarify.

Answer to Point 13: This sentence has been modified for clarity.

Point 14: “Neutrophils are minimally present in mammary tissue of healthy nulliparous mice, indicating they are part of a much larger immune program that plays a role in immune surveillance in the gland (42, 43).” This conclusion is unclear to me. Why does low numbers of neutrophils in the mammary gland indicate “..part of a much larger immune program..”. This conclusion seems too far-fetched.

Answer to Point 14: This sentence has been edited.

Point 15: “...associated with increased susceptibility and severe inflammatory responses to UTIs (47) The...”. Point missing.

Answer to Point 15: We apologize for this oversight. A period after every reference citation has been added to the revised manuscript.

Point 16: “Increased collagen content within the mammary gland has been described to have exponentially higher viscoelasticity, ...”. Is this association correct? Increased collagen is usually associated with less elasticity.

Answer to Point 16: From a mechanical standpoint, the “elasticity” of a tissue refers to its inherent physical properties, and is often referred to more plainly as a measure of tissue stiffness. Increased collagen is associated with increased stiffness, and thus an increased Young’s modulus/elevated elasticity within tissues. However, we recognize this may be counterintuitive to readers, and have thus amended the text for clarity.

Point 17: “Interestingly, studies analyzing extended antibiotic regimens for persistent UTIs did not show differences in the general incidence of breast cancer, suggesting that untreatable and persistent cases of UTI may correlate with an increased risk for cancer development (40).” Is

disagree with this conclusion. Please clarify why no changes in cancer incidence suggest increased risk for cancer.

Answer to Point 17: This sentence in the discussion has been modified for clarity.

Reviewer #3 (Remarks to the Author): with expertise in mammary gland, single cell genomics

This paper investigates how a urinary tract infections affect distal organs, focusing on tissue alterations in the mammary gland. The work utilises single cell RNA sequencing data to understand the transcriptional changes in mammary epithelial cells and reports upregulation of mechano-sensing and immune recruitment pathways. Using pregnancy models, the paper also reports delayed involution in the presence of UTIs. This is an interesting study and topic but requires further work to address outstanding questions as described in the comments below.

Major Points:

Point 1: Figure 1E: Summary graphs 1B and 1D are given to summarise the overall data from multiple glands and fields of view for ductal lumen area and collagen, respectively. Please could a similar summary graph be made to represent all of the images and data gathered from picosirius red staining in addition to the single representative screenshots shown?

Answer to Point 1: Quantitative analysis for Type I (Thick-red/yellow) and Type III (Thin – green) collagen fibers in the Picosirius red staining are now provided (**Fig. 1I-J**).

Point 2: NEC cluster naming is not very informative and difficult to follow, we recommend using more informative population names

Answer to Point 2: We have revised all scRNAseq panel for their content and lineage classification to address the reviewer's comments.

Point 3: UMAP shows 3 progenitor populations which cluster far away from each other. With the exception of some evidence of a small population of long-lived and primarily quiescent stem cells, many tracing studies now indicate that in the stable in-vivo adult mammary gland, basal and luminal lineages are self-maintained by unipotent progenitors. As mice are 7-9 weeks old, it would be unexpected to see bipotent progenitors, and more expected to see unipotent basal progenitors and ER-/PR- and ER+/PR+ luminal progenitors. Please could the authors review this data.

Answer to Point 3: We have revised all scRNAseq panel for their content and lineage classification to address the reviewer's comments.

Point 4: Supplemental figure 3a uses many non-canonical gene markers. Please could the authors provide references and/or justification for why these selected non-canonical gene markers are indicative of the cell type they are used to represent.

Answer to Point 4: The gene markers used to specify lineage identity were compiled from several bulk RNA sequencing and scRNAseq studies from mammary epithelial populations (Henry et al. 2021). Such markers were further validated using flow cytometry analysis, thus demonstrating their suitability to assign lineage identities to mammary epithelial cells. However, to address the reviewer's request, we have revised this panel (**Fig. 2B**) to included only commonly used markers of MEC lineages.

Point 5: Additionally, it appears some canonical cell type markers are assigned to incorrect cell populations. For example, immune cell markers CF52 and CD74 are categorised as bipotent progenitor markers. Please could the authors provide justification for this or update these as appropriate.

Answer to Point 5: To address the reviewer's request, we have revised this panel (Fig.2B) to only contain commonly used markers of MEC lineages.

Point 6: The YAP signature used is from a paper which used YAP overexpression in human mammary cell line MCF10As, or from mouse liver tissues or mouse immortalised fibroblasts (not mammary and also immortalised). Please could the authors provide justification as to why this a valid signature for YAP in the normal mouse mammary gland or update as appropriate.

Answer to Point 6: The YAP signature used in this study was derived from analyses of YAP/TAZ signaling in breast tumors (Cordenosi *et al.* 2011). We agree with the reviewer that the utilized signature was derived from MCF10A cells, however it has since been used by other groups to assess alterations to YAP signaling in mouse mammary epithelial cells (Zhang *et al.* 2009, Eyss *et al.* 2015). In order to address the reviewer's critique, we have edited our analysis, and included these additional published YAP-associated gene signatures in mammary cells/tumors, and reached similar conclusions as those presented in the original manuscript.

Point 7: Naming of populations is not in keeping with previous literature precedent. In particular, luminal ductal population lacks a reference or explanation and is not previously widely described. Please update where possible and appropriate to standard nomenclature.

Answer to Point 7: We have revised all scRNAseq panel for their content and lineage classification to address the reviewer's comments.

Point 8: Generally, in figure S3a some markers were assigned unexpected population names based on previous literature. For example:

- Markers for luminal ductal clusters and luminal progenitor cluster (NEC8) appear more like hormone sensing as they score highly for canonical markers such as ESR1 and AREG.
- Separate NEC9 clustering and higher expression of c-kit and Aldh1a3 suggests these may be hormone sensing progenitors.
- Aldh1a3 was assigned as a luminal alveolar marker but is canonically used as a luminal progenitor marker.

Please could the authors justify why these examples were assigned as stated or update as appropriate?

Answer to Point 8: We have revised all scRNAseq panel for their content and lineage classification to address the reviewer's comments.

Point 9: Please could the authors check marker gene set labels in figure S3a as these do not align with labels in S3b. For example, the population name for NEC8 being ascribed to LPs has no marker gene set shown in S3a?

Answer to Point 9: We have revised all scRNAseq panel for their content and lineage classification to address the reviewer's comments.

Point 10: Major: Figure 2 and Supplemental Figure 4 - Two obvious variables between NPN vs Pseudo without nursing- can a better comparator be made e.g. Natural pregnancy without nursing or Pseudo plus nursing (adoption)?

Answer to Point 10: In response to the critiques and concerns raised by the reviewer, we have removed datasets generated from pseudo-pregnant female mice, and replace it with analysis of naturally pregnant/lactating female mice. Using this system, we observed that post-lactation involution, UTI-bearing mice displayed signs of delayed involution, increased collagen deposition, and altered levels of circulatory factors. Neutrophil infiltration and adipocyte abundance were not altered in post-lactation involution, UTI-bearing mice (**Fig. 3, Fig. S4, Fig. S5, Fig. S8 and Fig. 5J-L**).

Point 11: Please could the authors clarify why pseudopregnancy was selected and outline what benefit this offers over the natural pregnancy alone model? Pseudopregnancy is not representative of something that happens often. Furthermore, the data shown (discussed further below) suggests these two pregnancy models vary quite significantly in their phenotype, which means they cannot be compared.

Answer to Point 11: In response to the critiques and concerns raised by the reviewer, we have removed datasets generated from pseudo-pregnant female mice, and replace it with analysis of naturally pregnant/lactating female mice.

Point 12: Pseudo control group without treatment needed to determine if UTI / TMS effect or pseudo pregnancy effect? (S4D suggests some of the UTI group have low bacteria counts. Are these wrongly classified?)

Answer to Point 12: In response to the critiques and concerns raised by the reviewer, we have removed datasets generated from pseudo-pregnant female mice, and replace it with analysis of naturally pregnant/lactating female mice.

Point 13: A control group given TMS alone might help justify that the UTI + treatment group are a valid control group by showing that additional treatment does not affect the mammary gland.

Answer to Point 13: To address the reviewer comment, we treated healthy animals that received transurethral injections of PBS with TMS alone, and we found that this approach did not induced alterations to mammary tissue (**Fig. S2D**).

Point 14: Evidence that the choice of mouse model might have a significant effect on the mammary gland phenotype being observed is indicated by the lack of correlation of the phenotype with the bacterial titre/degree of infection. For example, figures S4B-S4F shows that bacterial titre in UPEC mice are on average ~2x higher than pseudopregnancy mice. However, cytokeratin 5 (representing ductal structures) are more than double in pseudopregnancy mice relative to UPECs, and additionally, beta-casein retention is higher in pseudopregnancy mice relative to UPECs. The fact that the choice of mouse model significantly affects the phenotype may confound the cause and effect of the UTI, and more interestingly, implies that there may be additional model-specific mechanisms that affect the mammary gland phenotype which the authors could choose to pursue further.

Answer to Point 14: In response to the critiques and concerns raised by the reviewer, we have removed datasets generated from pseudo-pregnant female mice, and replace it with analysis of naturally pregnant/lactating female mice.

Point 15: Major :Figure 2J and 2I: • Please provide pseudo pregnancy PBS control or justification for absence of no baseline defined for pseudo pregnancy. Since natural pregnancy PBS control has similar collagen content to the pseudopregnancy with the UTI, it is unclear if the baseline for the pseudopregnancy vs natural pregnancy is different and it is difficult to determine whether pseudopregnancy + treatment is a suitable control.

Answer to Point 15: In response to the critiques and concerns raised by the reviewer, we have removed datasets generated from pseudo-pregnant female mice, and replace it with analysis of naturally pregnant/lactating female mice.

Point 16: The pseudopregnancy + UTI + treatment is a more complicated control and no control for antibiotic only is shown therefore it cannot be ruled out that TMS has an effect by itself other than curing the infection.

Answer to Point 16: In response to the critiques and concerns raised by the reviewer, we have removed datasets generated from pseudo-pregnant female mice, and replace it with analysis of naturally pregnant/lactating female mice.

Point 17: Could natural and pseudo pregnancy conditions be plotted on the same graph for easier and more robust statistical comparison?

Answer to Point 17: In response to the critiques and concerns raised by the reviewer, we have removed datasets generated from pseudo-pregnant female mice, and replace it with analysis of naturally pregnant/lactating female mice.

Point 18: Section 3: UTI induces neutrophil recruitment. Given it already well established that response to infection includes neutrophil recruitment, and given that another study on *Helicobacter Hepaticus* shows that the mammary gland recruits neutrophils in response to infection with a uropathogenic strain of *E. coli*, some further experiments may be required to make this section novel. For example, looking at neutrophil recruitment in other distal organs to see if this is a mammary specific response, or looking at other types of distal infection to see if this also has mammary specific neutrophil recruitment.

Answer to Point 18: We analyzed neutrophil migration (Myeloperoxidase+ IF) in other tissues, and we found increased Mpo+ cells in the liver, lung, pancreas, and spleen from UTI-bearing mice (Sup. 6E-F). Bone marrow and intestines from UTI-bearing mice did not increased Mpo+ cells compared to PBS controls. This finding supports that neutrophil infiltration is a broader consequence of host responses to UTI, rather than a mammary specific phenotype.

Point 19: The text references a 93% increase in Cd11bhi neutrophils in PBS and TMS groups. Was this an exact 93% increase Cd11bhi neutrophils in both PBS and TMS groups? Otherwise, we would request the authors report the PBS and TMS respective values.

Answer to Point 19: We have amended the manuscript to report independent values for each comparison made wherever applicable.

Point 20: TIMP1 increases circulatory factors which increases neutrophil expansion, however it is currently unclear if TIMP1 induces changes directly or indirectly via the circulatory factors. Please could the authors clarify if and how additional circulatory factors were ruled out as causes of the increase in mammary resident neutrophils and alteration to the ECM and ductal structures.

If not already investigated, further additional work to rule out circulating factors as a cause would be insightful.

Answer to Point 20: In order to address the reviewer comment, we treated nulliparous UTI-bearing mice with CSF3 (G-CSF) neutralizing antibodies or IgG control. We found that such approach decreased the abundance of collagen deposition in UTI-bearing mice, with no alteration to the severity of infection, adipocyte content, nor the abundance of mammary infiltration or circulatory neutrophils (**Fig. 4F-I** and **Fig. S6I-N**). These results agree with the role of anti-CSF3 treatment in alleviating collagen-induced arthritis and inducing expression of collagen related genes and fibroblast populations, and suggests that multiple factors may play a role in controlling UTI-induced mammary tissue alterations.

Point 21: We recommend the same in-depth analysis pre- and post- anti-TIMP1 administration to understand if treatment reverses all the phenotypes previously reported in the paper observed on infection. If there is not a complete return to the original phenotype, it might be of interest to explore the consequences or causes of that further.

Answer to Point 21: Treatment of nulliparous UTI-bearing mice with TIMP1 neutralizing antibodies reverted all of UTI-induced alterations to the mammary tissue, supporting its specific action on mammary gland tissue health. With the attempt to further address the reviewer critique, we treated post-lactation involution, UTI-bearing mice with either control IgG or TIMP1 neutralizing antibodies at the onset of involution. Our results indicate that such treatment did not alter signs of delayed involution, but did ameliorate collagen accumulation, further implicating TIMP1 as the major regulator of collagen accumulation in the mammary tissue of UTI-bearing mice (**Fig. 5K-L** and **Fig. S8**).

Point 22: Frequent switching and comparison between natural and pseudo pregnancy is quite difficult to follow. The acronyms used are not easy to follow. Please review whether more self explanatory acronyms could be used.

Answer to Point 22: In response to the critiques and concerns raised by the reviewer, we have removed datasets generated from pseudo-pregnant female mice, and replace it with analysis of naturally pregnant/lactating female mice

Point 23: Figure legends could be much more informative and should define the acronyms used in each legend to more easily stand alone from text.

Answer to Point 23: We have edited all figure legends to include more technical details and acronym definitions.

Point 24: Discussion is slightly speculative (Lines 315-318) and in places difficult to follow (Lines 331-334). Please make more focused and data driven.

Answer to Point 24: We have edited the discussion to improve clarity and accuracy.

Point 25: Different routes of administration: Bladder inoculation vs transurethral administration. Can a justification be provided for this?

Answer to Point 25: We induce UTIs by inoculating *E. coli* directly into the bladder, using a transurethral route. We have clarified this information on the methods section.

Point 26: Please could the authors provide details for recombinant TIMP1 protein (rTIMP1) along with details on how the dose concentration & treatment duration were selected? Concentration used was much higher than concentration measured.

Answer to Point 26: To improve the clarity and focus of our manuscript, because no effect was observed with the mentioned experiment, we have removed the data from the revised manuscript.

Point 27: Figure 1F & Supplemental Figure 3: Data suggests potential doublet populations within the data set. Please could the authors provide justification for why stromal cells were excluded using MACS beads and not FACS? Additionally, please could they provide details of any steps (e.g. flow cytometry) performed to validate this as an effective method for excluding lineage positive cells?

Answer to Point 27: All of the scRNA-seq analysis performed in this study utilized integrated data which is not compatible with doublet separation. We have also clarified all QC steps to ensure reliable data utilization on the methods section.

In addition, the information about the removal of stromal cells was an error, thus all datasets utilized in this study have both epithelial (MECs), and non-epithelial cells (immune, stroma, etc). In fact, we have included new analysis indicating that specific populations of fibroblasts are present in the mammary tissue of nulliparous UTI-bearing mice, and may represent the source of increased collagen deposition (**Fig. 2D-I**, and **Fig. S5D-H**). Such population of fibroblasts was not present in the mammary tissue of involuting, UTI-bearing mice (**Fig. 3I-K**, and **Fig. S5D-H**).

In regards to the preparation of scRNA-seq datasets, whole mammary tissue was processed and submitted for library preparation, and only RBCs were removed utilizing RBC lysis buffer, as described in the methods section.

Point 28: The inclusion of flow cytometry data generated from samples post MACS bead filtering would be beneficial to demonstrate this as an effective approach to exclude lineage positive cells as would the inclusion of doublet scoring and exclusion in scRNA-seq QC pipeline.

Answer to Point 28. We have updated the methods section to clarify all steps of data QC performed. MACS beads cell exclusion was not utilized, and this information was updated on the methods section.

Point 29: Minor: Data processing and analysis for scRNA-seq: Please provide information on the type of batch correction or normalisation used along with a justification for why this was appropriate.

Answer to Point 29: We have updated the methods section to clarify all steps of data QC, filtering and analysis performed.

Point 30: Minor: Figure S5: Please provide information and justification on why PEC was batch corrected, normalised and clustered separately to the NECs?

Answer to Point 30: Nulliparous (NP) and Post-lactation involution (PLI) datasets were handled separately in order to enable examination of the influence of UTIs on the mammary cells, independently and without convolution by parity status.

Point 31: Please update PEC# to more informative population names.

Answer to Point 31: We have revised all scRNAseq panel for their content and lineage classification to address the reviewer's comments.

Point 32: Please provide justification as to why all of the UTI groups in this figure are pseudopregnancy mice and why these were used instead of natural pregnancy group or a combination of the two.

Answer to Point 32: In response to the critiques and concerns raised by the reviewer, we have removed datasets generated from pseudo-pregnant female mice, and replace it with analysis of naturally pregnant/lactating female mice.

Point 33: Minor: Figure S5C. We recommend using gene signatures composed of multiple genes to represent pathways rather than single genes selected to represent whole pathways as this is likely to be more representative.

Answer to Point 33: We have replaced this analysis, with the representation of genes associated with mammary gland involution post-pregnancy (Stein *et al.* 2004).

Point 34: We recommend plotting the average expression of individual mice rather than plotting individual cells and repeating P-values with individual mouse average expression. (Statistically comparing the difference in expression of single cells assumes each cell can be considered as completely independent events. However, bias is introduced by cells coming from the same mice and being prepared at the same time.

Answer to Point 34: All of the scRNAseq libraries utilized in this study were generated by pulling mammary tissue (all pairs) from 2 animals per condition, therefore we cannot plot individual mouse per condition. All of the cellular analysis utilized in this study were re-analyzed by the CSHL biostatistician, and now co-author in this manuscript, Dr. Taehoon Ha, Briefly, datasets were assessed for normality and log transformed, to adjust to a normal distribution. Statistical significance testing was carried out using a linear mixed effects model to compare conditions while accounting for both, fixed effects (group) and random effects (data points nested within the same biological replicate). Additionally, all bar & scatter plots for image-based analyses have been replaced with violin plots to better reflect the data distribution across animals. With this approach, biases introduced by cells coming from the same animals, and being prepared at the same time, are minimized.

Point 35: Y axis labelling- requires clarification as to units.

Answer to Point 35: We have updated the Y axis of all plots to clarify the units.

REVIEWER COMMENTS

Reviewer #1 (Remarks to the Author):

The authors addressed my comments in a satisfactory way.

Reviewer #2 (Remarks to the Author):

Dear Authors,

thank you very much for addressing my questions. However, my major concern was not sufficiently addressed.

My major concern:

Point 1: Why was the statistics be performed ROI- and not sample-based? In case e.g. 50 ducts per gland were analyzed, one mean or median should be used as „n”, and not a value of “50”. This concern refers to several histology-based figures. This might significantly change most of the statements in the manuscript.

Your response:

Answer to Point 1: The mammary gland has considerable heterogeneity in tissue size and composition, including the distribution of collagen deposits. Therefore, for the imaging analyses in this study we employed a hierarchical/spatial repeated measures approach commonly used in mammary histological assessment (O’Brien et al. 2010, Lyons et al. 2011, Maller et al. 2013, Guo et al. 2022). In reporting a singular mean/median, information about observational variability is lost and precision of data reporting is lowered (Lazic 2021).

In order to fully address the reviewer’s concern, all of datasets were re-analyzed by the CSHL biostatistician, and now co-author in this manuscript, Dr. Taehoon Ha, Briefly, datasets were assessed for normality and log transformed, to adjust to a normal distribution.

Statistical significance testing was carried out using a linear mixed effects model to compare conditions while accounting for both, fixed effects (group) and random effects (data points

nested within the same biological replicate).

A detailed description of the biostatistical analyses has been included in the supplementary methods. Additionally, all bar & scatter plots for image-based analyses have been replaced with violin plots to better reflect the data distribution and p-values have been amended, as described above, in the revised manuscript.

My comment to your response:

Unfortunately, I cannot make a final assessment with the information you provide. You need to write more precisely where the information about the statistics in the Supplementary Methods can be found in the revised version. In the "Supplementary Experimental Procedures" and also in the merged pdf file that was made available to me, I cannot find any information about the "linear mixed effects model" mentioned in the rebuttal and the explanations about the statistics. I would like to understand these in order to be able to submit my evaluation. Where can I find this information? Please provide clear information and publications on why multiple ROIs from the same sample can be used for a statistic. You are artificially increasing the sample size here and potentially achieving stronger significance than if only biological samples were used as the sample size.

Furthermore, if you specify references in the rebuttal and these references are not contained in the manuscript, then information about DOI or PMID would be necessary or a reference list at the end of the rebuttal. I cannot find the corresponding publications in this form and therefore cannot understand your argumentation: Your answer in the rebuttal: "Therefore, for the imaging analyses in this study we employed a hierarchical/spatial repeated measures approach commonly used in mammary histological assessment (O'Brien et al. 2010, Lyons et al. 2011, Maller et al. 2013, Guo et al. 2022)." I need further information about the publications.

Best regards and thank you for answering my questions.

Reviewer #3 (Remarks to the Author):

The authors have addressed most of my concerns and the manuscript is improved. I have no further comments.

Responses to Reviewer Comments

We would like to thank the reviewers for acknowledging the improvement we made on our manuscript, and for allowing an additional opportunity to further clarify some of the analytical approaches used in our study. Please find below our responses (in blue font) to reviewer' concerns.

Reviewer #1 (Remarks to the Author): with expertise in mammary genome, mammary glands and pregnancy

Reviewer Comment: The authors addressed my comments in a satisfactory way.

Response to Reviewer Comment: We would like to thank the reviewer for endorsing our efforts to improve the clarity of our studies.

Reviewer #2 (Remarks to the Author): with expertise in neutrophils and urinary infections

Dear Authors, thank you very much for addressing my questions. However, my major concern was not sufficiently addressed.

Reviewer Comment: My major concern: Point 1: Why was the statistics be performed ROI- and not sample-based? In case e.g. 50 ducts per gland were analyzed, one mean or median should be used as „n”, and not a value of “50”. This concern refers to several histology-based figures. This might significantly change most of the statements in the manuscript.

Your response: Answer to Point 1: The mammary gland has considerable heterogeneity in tissue size and composition, including the distribution of collagen deposits. Therefore, for the imaging analyses in this study we employed a hierarchical/spatial repeated measures approach commonly used in mammary histological assessment (O'Brien et al. 2010, Lyons et al. 2011, Maller et al. 2013, Guo et al. 2022). In reporting a singular mean/median, information about observational variability is lost and precision of data reporting is lowered (Lazic 2021). In order to fully address the reviewer's concern, all of datasets were re-analyzed by the CSHL biostatistician, and now co-author in this manuscript, Dr. Taehoon Ha, Briefly, datasets were assessed for normality and log transformed, to adjust to a normal distribution. Statistical significance testing was carried out using a linear mixed effects model to compare conditions while accounting for both, fixed effects (group) and random effects (data points nested within the same biological replicate). A detailed description of the biostatistical analyses has been included in the supplementary methods. Additionally, all bar & scatter plots for image-based analyses have been replaced with violin plots to better reflect the data distribution and p-values have been amended, as described above, in the revised manuscript.

My comment to your response: Unfortunately, I cannot make a final assessment with the information you provide. You need to write more precisely where the information about the statistics in the Supplementary Methods can be found in the revised version. In the "Supplementary Experimental Procedures" and also in the merged pdf file that was made available to me, I cannot find any information about the "linear mixed effects model" mentioned in the rebuttal and the explanations about the statistics. I would like to understand these in order to be able to submit my evaluation. Where can I find this information? Please provide clear information and publications on why multiple ROIs from the same sample can be used for a statistic. You are artificially increasing the sample size here and potentially achieving stronger

significance than if only biological samples were used as the sample size.

Furthermore, if you specify references in the rebuttal and these references are not contained in the manuscript, then information about DOI or PMID would be necessary or a reference list at the end of the rebuttal. I cannot find the corresponding publications in this form and therefore cannot understand your argumentation: Your answer in the rebuttal: "Therefore, for the imaging analyses in this study we employed a hierarchical/spatial repeated measures approach commonly used in mammary histological assessment (O'Brien et al. 2010, Lyons et al. 2011, Maller et al. 2013, Guo et al. 2022)." I need further information about the publications. Best regards and thank you for answering my questions.

Response to Reviewer Comment: We would like to thank the reviewer for indicating the incomplete description of statistical approaches in our manuscript. We deeply apologize for this oversight. This information is now included on the Supplementary Methods section, including the citation of above indicated studies that utilize multiple ROIs to investigate mammary tissue architecture and cellular compositions. All alterations to Supplementary Methods section are indicated in red.

O'Brien et. al. 2010 - <https://www.ncbi.nlm.nih.gov/pmc/articles/PMC2832146/>

Lyons et. al. 2011 - <https://pubmed.ncbi.nlm.nih.gov/21822285/>

Lloyd-Lewis et. al. 2017 - https://experiments.springernature.com/articles/10.1007/978-1-4939-6475-8_7

Liu et al. 2017 - <https://pubmed.ncbi.nlm.nih.gov/28836218/>

Wormsbaecher et. al. 2020 - <https://breast-cancer-research.biomedcentral.com/articles/10.1186/s13058-020-01275-w>

Monks et al. 2022 - <https://www.frontiersin.org/articles/10.3389/fcell.2022.958566/full>

We would also like to address Reviewer concern that our approach "*artificially increasing the sample size here and potentially achieving stronger significance than if only biological samples were used as the sample size*". The selection of an approach that utilizes multiple ROIs to investigate mammary tissue alterations ensures analysis across multiple regions, biological replicates, and conditions, while avoiding biases towards specific tissue regions. Such strategy, paired with a linear mixed-effect model, is suitable for multiple measurements from the same unit (in this case, multiple ROIs from the same gland), and mitigates the risk of overestimating the significance of a given analysis.

With the attempt to demonstrate the robustness of our analytical approach, and to address the reviewer concern, we conducted an analysis where the average ductal lumen area per sample was calculated and plotted (Figure R1). This approach yielded the same result and statistical power as when multiple ROIs are plotted individually (shown on **Fig. 1C**). However, such approach does not illustrate variations within the same tissue, which we believe it is an important information that must be included, given the heterogenous nature of mammary tissue. Therefore, this result demonstrates that our analytical approach, considering multiple ROIs, does not overestimate the significance or magnitude of the evaluated mammary tissue abnormalities.

Figure R1. Quantification of ductal lumen area in mammary tissue from PBS control mice (n=9), from UTI-bearing mice (n=8) and from UTI-bearing mice treated with TMS (n=5) at 2 weeks post-infection (p.i.). Multiple mammary ducts (50) were quantified per gland and then averaged by samples and plotted individually. ** $p \leq 0.01$; *** $p \leq 0.001$.

We hope that the current clarification and improved description of methods are sufficient to demonstrate that our research was performed with rigor, and that our conclusions were based on properly controlled and analyzed experimental approaches.

Reviewer #3 (Remarks to the Author): with expertise in mammary gland, single cell genomics

Reviewer Comment: The authors have addressed most of my concerns and the manuscript is improved. I have no further comments.

Response to Reviewer Comment: We would like to thank the reviewer for endorsing our efforts to improve the clarity of our studies.

REVIEWERS' COMMENTS

Reviewer #2 (Remarks to the Author):

Dear Authors,

thank you very much for having addressed my comments in a satisfactory way and included information on the statistics in the supplementary section.

Best regards and congratulations on this study